# Model predictive game control for personalized and targeted interactive assistance
Abdelwaheb Hafs [1]✉, Anaïs Farr [1], Dorian Verdel [2], Olivier Bruneau[3], Etienne Burdet [2] & Bastien Berret [1]✉

Contact robots are increasingly used to assist humans in physical training and manufacturing tasks. However, their effectiveness is currently limited as control methods focus on system performance without considering the upcoming human user's control. Here, we present a differential game-based controller for contact robots ensuring optimal interaction by predicting human motor control over their finite planning horizon. Using this model-predictive game (MPG) controller, we investigated human-robot co-adaptation in experiments, demonstrating that: (a) MPG interaction remains stable while reducing human effort; (b) the robot adapts to humans, identifying time-consistent individual interaction behaviors; (c) humans adapt to the robot, and their behavior can be modulated through an assistance meta-parameter adjusting the robot's propensity to minimize human effort. These findings indicate that humans understand and adapt to a partner's control strategy aligning with game theory principles. Furthermore, the assistance meta-parameter's ability to guide humans toward specific interaction behaviors enables versatile robot-assisted systems for physical training and rehabilitation.

Robots working in physical contact with a human are increasingly being used for applications ranging from sports training and physical rehabilitation[1], to co-manipulation of large or heavy objects in construction and manufacturing[2]. These activities have traditionally been performed with another human, such as a physical trainer, therapist or co-worker, who arguably: (a) interacts optimally with the partner during movement by continuously adjusting to their motor plan, (b) adapts to the partner's dynamics, e.g., by sensing whether a patient can actively move their limb or requires greater assistance, and (c) promotes different interaction behaviors in the partner, such as relaxing, increasing strength, or learning a new skill by initially guiding the motion and progressively letting the partner take the lead. However, current approaches to movement assistance based on improvement of performance metrics using adaptive/iterative control[3–7] or human-in-the-loop optimization[8–10] cannot achieve optimal interaction with the human user, as they do not account for their motor planning capacities[11].

To develop a robotic partner with above properties, the concept paper[12] proposed modeling a contact robot and its human user as two agents with their own actuation, sensing and motion planning ability. In this context, differential games (DG) theory was suggested as a promising framework to flexibly adjust the interaction paradigm, with possibilities ranging from competition to cooperation (assistance). Li et al.[13] introduced an adaptive

DG algorithm that identifies the human user's control dynamics and demonstrated the superiority of this optimal interactive behavior over independent control. However, both this and subsequent DG algorithms[14,15] essentially rely on an infinite planning horizon, while human motor planning considers a limited horizon[16]. Furthermore[13–15], implemented only a single interaction behavior, lacking the versatility to inducing diverse behaviors in the human partner.

Here we develop a *model predictive game* (MPG) controller for a contact robot that continuously infers the human motor control on a finite horizon[17], i.e., with properties (a) and (b). To address (c), we incorporate a homotopy-based mechanism with an assistance meta-parameter that regulates the robot's internal objectives in terms of human effort minimization. While previous studies successfully implemented DG-based assistance controllers inducing stable interaction between human and robot agent[13,17], co-adaptation between the two agents has yet to be more thoroughly investigated. Would users actively contribute to the movement even when the robot could carry out the task alone? And can MPG ensure a stable human-robot interaction with users having different control dynamics? To address these questions and investigate the human-robot co-adaptation, we conducted systematic experiments with 30 participants using a wrist exoskeleton equipped with our MPG algorithm.

[1]Université Paris-Saclay, Inria, CIAMS, Gif-sur-Yvette, France. [2]Imperial College of Science, Technology and Medicine, London, UK. [3]LURPA, ENS Paris-Saclay, Université Paris-Saclay, Gif-sur-Yvette, France. ✉e-mail: abdelwaheb.hafs@universite-paris-saclay.fr; bastien.berret@universite-paris-saclay.fr

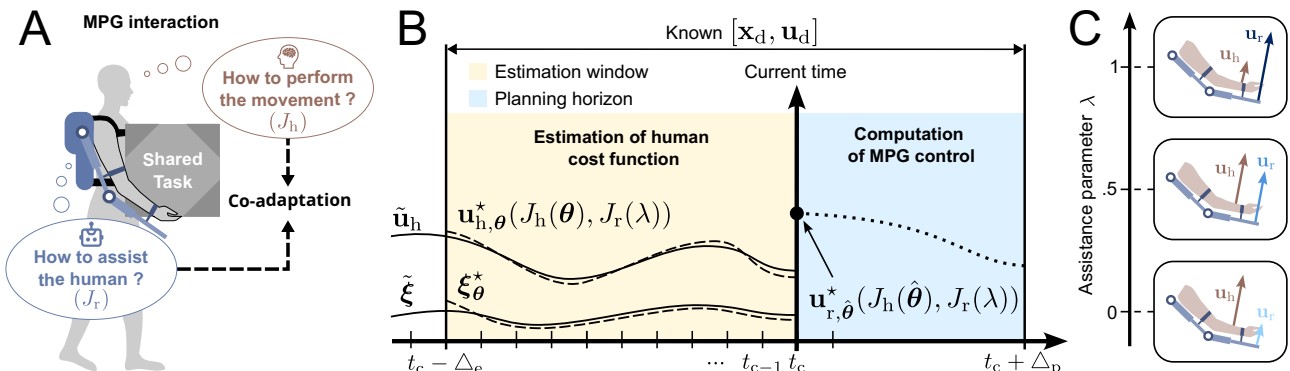

**Fig. 1 | Model predictive game (MPG) control concept. A** The interaction control behavior of the robot and human model are defined by their respective cost functions $J_r$ and $J_h$, enabling their co-adaptation. **B** Estimation process over time using a receding horizon approach with current time $t_c$. The human cost parameters $\boldsymbol{\theta}$ are adjusted to fit past interaction data (e.g., the measured human inputs $\tilde{\mathbf{u}}_h$) on the estimation window. Once an estimate $\hat{\boldsymbol{\theta}}$ is obtained, the control law can be computed by solving a single dyadic affine-quadratic differential game (DG), and the first value $\mathbf{u}^\star_{r,\hat{\boldsymbol{\theta}}}$ is used as motor command to the robot. **C** A range of interaction behaviors can be achieved by adjusting the assistance level $\lambda \in [0, 1]$. This parameter balances the contributions of human and robot motor commands, $\mathbf{u}_h$ and $\mathbf{u}_r$, respectively.

Our MPG framework assumes that the human and robot engage in a non-cooperative DG where each agent seeks to minimize their cost function in fixed time without explicit collaboration or binding agreement (Fig. 1A). It formulates a finite-horizon affine-quadratic DG that enables closed-form solutions for a feedback Nash equilibrium defining the optimal interactive strategy[18]. To formulate the cost functions, we assume that (i) the human user's motor control minimizes performance error and effort[19–23], while (ii) the robot minimizes the human effort in addition to own error and effort. Importantly, this robot cost function considering both human and robot efforts can theoretically be used to tune assistance (Fig. 1C) and promote different interaction behaviors in the human user.

A key challenge for the MPG framework lies in reliably and continuously estimating the human cost function parameters during interaction. This estimation is formulated as a finite-horizon inverse problem, and solved using a bi-level optimization approach[24,25]. The human model cost parameters that best fit past interaction data are identified to compute the optimal interaction command for the robot (Fig. 1B), leveraging a predictive model of the upcoming human motor plan.

## Results

### Interaction control as differential game on a receding finite horizon

Let the human-robot interaction dynamics be described by $\dot{\mathbf{x}}(t) = \mathbf{f}(\mathbf{x}(t), \mathbf{u}(t), t)$, where $\mathbf{f}$ is a smooth function of the common state vector $\mathbf{x} \in \mathbb{R}^n$ and the command vector $\mathbf{u} \in \mathbb{R}^m$ merging the human ($\mathbf{u}_h$) and robot ($\mathbf{u}_r$) inputs. We assume that the planned trajectory of the interaction dynamics ($\mathbf{x}_d$) is known on a *finite planning horizon* $\Delta_p > 0$ and that their control policy remains to be identified[17]. This situation is encountered in trajectory-tracking tasks used for robot-aided rehabilitation[26], while methods to infer upcoming targets or trajectories are considered elsewhere, e.g.,[27,28]. Assuming that the desired control input $\mathbf{u}_d$ can be calculated using inverse dynamics from $\mathbf{x}_d$ (e.g., when $\mathbf{f}$ represents the robot's rigid body dynamics and all degrees of freedom can be directly controlled), the dynamics can then be linearized around ($\mathbf{x}_d, \mathbf{u}_d$):

$$\dot{\boldsymbol{\xi}}(t) = \mathbf{A}(t)\boldsymbol{\xi}(t) + \mathbf{B}(t)[\mathbf{u}_r(t) + \mathbf{u}_h(t)] + \mathbf{c}(t), \; \boldsymbol{\xi} = \mathbf{x} - \mathbf{x}_d, \quad (1)$$

$$\mathbf{A}(t) \triangleq \frac{\partial \mathbf{f}}{\partial \mathbf{x}}|_{(\mathbf{x}_d, \mathbf{u}_d)}, \; \mathbf{B}(t) \triangleq \frac{\partial \mathbf{f}}{\partial \mathbf{u}}|_{(\mathbf{x}_d, \mathbf{u}_d)}, \; \mathbf{c}(t) = -\mathbf{B}(t)\mathbf{u}_d(t).$$

This affine system of equations describes how the human and robot inputs affect $\boldsymbol{\xi}$, the deviations from the planned trajectory that the agents will attempt to minimize. In practice, this linearization is valid when deviations from the desired trajectory remain small. This assumption is plausible in a variety of human-robot interaction scenarios, either because (i) both agents are provided with the desired trajectory and instructed to track it, e.g., during a rehabilitation protocol; or (ii) because the robot is able to extract an accurate estimation of the desired trajectory of the human, e.g., during an industrial task. In general, the method requires the knowledge of both a reference state and control trajectory, a process which may require leveraging other methods (e.g.,[29]), but this is out of the scope of the present paper and left for future work. A more general formulation is presented in Supplementary Note 1, and an analysis of the advantages of the finite-horizon formulation over the infinite-horizon one is provided in Supplementary Note 2.

We consider that both the human and robot agents aim at minimizing a quadratic tracking error over a time horizon $[\tau, \tau + \Delta_p]$. As humans tend to minimize effort during their actions[22], we set that the human motor control minimizes the following cost function:

$$J_h = \frac{1}{2}\boldsymbol{\xi}(\tau_f)^\top \mathbf{Q}_h^f \boldsymbol{\xi}(\tau_f) + \frac{1}{2}\int_\tau^{\tau_f} \left[\boldsymbol{\xi}(t)^\top \mathbf{Q}_h \boldsymbol{\xi}(t) + \mathbf{u}_h(t)^\top \mathbf{R}_h \mathbf{u}_h(t)\right] dt, \; \tau_f = \tau + \Delta_p.$$

$$(2)$$

To let the robot fulfill an assistive function, we assume that its control minimizes human effort in addition to its own tracking error and effort:

$$J_r = \frac{1}{2}\boldsymbol{\xi}(\tau_f)^\top \mathbf{Q}_r^f \boldsymbol{\xi}(\tau_f) + \frac{1}{2}\int_\tau^{\tau_f} \left[(\boldsymbol{\xi}(t)^\top \mathbf{Q}_r \boldsymbol{\xi}(t) + \mathbf{u}_r(t)^\top \mathbf{R}_r \mathbf{u}_r(t) + \mathbf{u}_h(t)^\top \mathbf{R}_{rh} \mathbf{u}_h(t)\right] dt.$$

$$(3)$$

In above equations, $\mathbf{Q}_r, \mathbf{Q}_r^f, \mathbf{Q}_h, \mathbf{Q}_h^f \in \mathbb{R}^{n \times n}$ are positive semi-definite matrices weighting task error minimization, which increase the amount of effort that agents can expand to reduce tracking errors, while $\mathbf{R}_r \in \mathbb{R}^{m \times m}$ and $\mathbf{R}_h \in \mathbb{R}^{m \times m}$ are positive definite matrices weighting effort minimization, resulting in a compromise between tracking accuracy and effort expenditure. The positive semi-definite matrix $\mathbf{R}_{rh} \in \mathbb{R}^{m \times m}$ reflects the extent to which the robot aims to assist the human partner. Note that the human cost function does not include a $\mathbf{R}_{hr} \in \mathbb{R}^{m \times m}$ term as there is no reason to assume that humans would attempt to minimize the partner's effort in our robotic assistance setup. However, such a term could appear during interactions with a social component, such as between humans or with social robots.

We introduce a meta-parameter $\lambda$ to modulate the robot's assistance level, by using a prefixed matrix $\overline{\mathbf{R}}$ and writing the effort-related costs of the

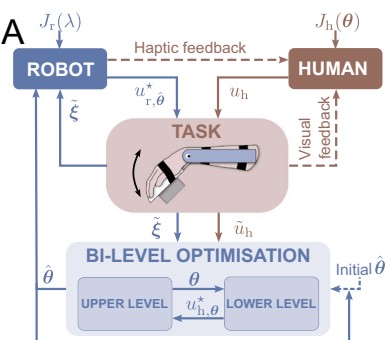
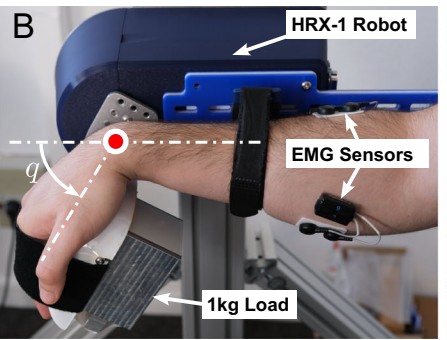
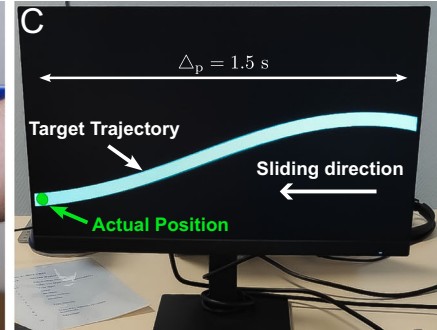

**Fig. 2 | Control scheme and experimental task with 1-DOF exoskeleton.**
**A** Control with online estimation of the human model. Haptic and visual feedback are provided to the human participant throughout the task. $J_h(\theta)$ and $J_r(\lambda)$ represent the human and robot cost functions, respectively, $u_h$ and $u^\star_{r,\theta}$ their applied torque during movement. $\tilde{u}_h$ is the measured human torque, $\tilde{\xi}$ the state deviation, and $(\hat{\theta})$ the human cost parameter estimated via bi-level optimization. **B** The human and the

robot (with 1 kg load) are mechanically connected to perform the tracking task, with EMG activity recorded from a wrist antagonist muscle pair. **C** The target trajectory is displayed on a screen over a finite horizon $\Delta_p$. The actual robot position is represented by a green disk when inside the trajectory and a red disk when outside it. The disk moves only vertically, while the target trajectory slides toward it at a constant velocity.

robot as:

$$\mathbf{R}_r = \varepsilon_r \mathbf{I} + (1-\lambda)\overline{\mathbf{R}}, \ \mathbf{R}_{rh} = \varepsilon_h \mathbf{I} + \lambda\overline{\mathbf{R}}, \ 0 \le \lambda \le 1, \tag{4}$$

where $\varepsilon_h, \ \varepsilon_r > 0$ are set to ensure well-posedness and numerical stability for any $\lambda$. To clarify how the parameter $\lambda$ can be used to adjust the degree of assistance, we write the robot's cost function as:

$$J_r(\lambda) = \frac{1}{2}\int_\tau^{\tau_f}[(1-\lambda)\mathbf{u}_r^\top\overline{\mathbf{R}}\mathbf{u}_r + \lambda\mathbf{u}_h^\top\overline{\mathbf{R}}\mathbf{u}_h + S]\,dt, \ S = \boldsymbol{\xi}^\top\mathbf{Q}_r\boldsymbol{\xi} + \varepsilon_r\mathbf{u}_r^\top\mathbf{u}_r + \varepsilon_h\mathbf{u}_h^\top\mathbf{u}_h. \tag{5}$$

We see that assistance is minimal for $\lambda = 0$, maximal for $\lambda = 1$, and tunable in between. Notably, the cost function with $\lambda = 0$ used in Ref. 13,15 and $\lambda = 0.5$ in Ref. 14 restricted the robot's behavior to a single interaction behavior. In contrast, by tuning the assistance parameter $0 \le \lambda \le 1$, the robot can achieve a broad range of interaction behaviors.

((1))–((3)) define a dyadic affine-quadratic DG of prefixed duration $\Delta_p$. Under the closed-loop perfect state information pattern, Corollary 6.5 in Ref. 18 (page 323) states that a feedback Nash equilibrium strategy exists. The optimal control for each agent can be obtained by starting with $\mathbf{P}_i(\tau_f) = \mathbf{Q}_i^f, \ \boldsymbol{\alpha}_i(\tau_f) = \mathbf{0}, i \in \{r, h\}$ and solving the following set of ordinary differential equations with unknowns matrices and vectors $\mathbf{P}_h(t), \boldsymbol{\alpha}_h(t), \mathbf{P}_r(t), \boldsymbol{\alpha}_r(t)$ backward in time $t$ for $\tau_f > t > \tau$ :The optimal control for each agent can be obtained by integrating a set of ordinary differential equations with unknowns matrices and vectors $\mathbf{P}_h(t), \boldsymbol{\alpha}_h(t), \mathbf{P}_r(t), \boldsymbol{\alpha}_r(t)$ backward in time. First, we initialize the value of these matrices at the end of the planning horizon $\mathbf{P}_i(\tau_f) = \mathbf{Q}_i^f, \ \boldsymbol{\alpha}_i(\tau_f) = \mathbf{0}, i \in \{r, h\}$. Then the following equations are iteratively integrated backwards in time, which is by starting from $t = \tau_f$ and finishing with $t = \tau$ the current time, to obtain the time evolution of the unknown matrices over the planning horizon $\Delta_p$:

$$-\dot{\mathbf{P}}_h = \mathbf{F}^\top\mathbf{P}_h + \mathbf{P}_h\mathbf{F} + \mathbf{Q}_h + \mathbf{P}_h\mathbf{B}\mathbf{R}_h^{-1}\mathbf{B}^\top\mathbf{P}_h$$
$$-\dot{\boldsymbol{\alpha}}_h = \mathbf{F}^\top\boldsymbol{\alpha}_h + \mathbf{P}_h\boldsymbol{\beta} + \mathbf{P}_h\mathbf{B}\mathbf{R}_h^{-1}\mathbf{B}^\top\boldsymbol{\alpha}_h$$
$$-\dot{\mathbf{P}}_r = \mathbf{F}^\top\mathbf{P}_r + \mathbf{P}_r\mathbf{F} + \mathbf{Q}_r + \mathbf{P}_r\mathbf{B}\mathbf{R}_r^{-1}\mathbf{B}^\top\mathbf{P}_r + \mathbf{P}_h\mathbf{B}\mathbf{R}_h^{-1}\mathbf{R}_{rh}\mathbf{R}_h^{-1}\mathbf{B}^\top\mathbf{P}_h \tag{6}$$
$$-\dot{\boldsymbol{\alpha}}_r = \mathbf{F}^\top\boldsymbol{\alpha}_r + \mathbf{P}_r\boldsymbol{\beta} + \mathbf{P}_r\mathbf{B}\mathbf{R}_r^{-1}\mathbf{B}^\top\boldsymbol{\alpha}_r + \mathbf{P}_h\mathbf{B}\mathbf{R}_h^{-1}\mathbf{R}_{rh}\mathbf{R}_h^{-1}\mathbf{B}^\top\boldsymbol{\alpha}_h$$
$$\mathbf{F} \triangleq \mathbf{A} - \sum_{i\in\{r,h\}}\mathbf{B}\mathbf{R}_i^{-1}\mathbf{B}^\top\mathbf{P}_i, \ \boldsymbol{\beta} \triangleq \mathbf{c} - \sum_{i\in\{r,h\}}\mathbf{B}\mathbf{R}_i^{-1}\mathbf{B}^\top\boldsymbol{\alpha}_i.$$

The control law of the robot and human model agents then arise as a combination of feedforward and feedback terms:

$$\mathbf{u}_i(t, \boldsymbol{\xi}(t)) = -\mathbf{R}_i^{-1}\mathbf{B}(t)^\top[\boldsymbol{\alpha}_i(t) + \mathbf{P}_i(t)\boldsymbol{\xi}(t)], \ i \in \{r, h\}, \ t \in [\tau, \tau + \Delta_p]. \tag{7}$$

These optimal control laws implement a Nash equilibrium and solve the dyadic affine-quadratic DG problem on the planning horizon of duration $\Delta_p$. In the MPG framework, only the first value of the optimal robot command at time $\tau$ is used for control. Note that the controller can be applied for multiple timesteps if needed.

**Inverse DG for human cost estimation**
As the robot does not a priori know the human cost function, the matrices $\mathbf{Q}_h, \mathbf{Q}_h^f, \mathbf{R}_h$ are identified using a bi-level optimization approach combining solving of the affine-quadratic DG problem given a human cost, and best fitting of human-robot interaction data on an estimation window (Fig. 2A). For this purpose, we use diagonal matrices $\mathbf{Q}_h, \mathbf{Q}_h^f, \mathbf{R}_h$, which correspond to the assumptions that humans (i) correct errors in position and velocity independently along each task dimension, and (ii) minimize control costs independently for each task dimension. Both these assumptions are plausible in humans and have allowed seamless human-robot interactions using infinite horizon DG control[13]. We then collect the $p = 2n + m$ human cost parameters in a vector $\boldsymbol{\theta} \in \mathbb{R}^p$, which will be identified in the *estimation window* $[t_c - \Delta_e, t_c]$ when $t_c > \Delta_e > 0$, where $t_c$ is the *current time* in the task.

The *lower-level problem* is to compute the optimal commands for the human and robot agents and the corresponding trajectory, by minimizing the robot cost in eq. (3) and the human model cost in eq. (2) parametrized by $\boldsymbol{\theta}$, i.e., $J_h(\boldsymbol{\theta})$. The Nash equilibrium solution is obtained by solving eq. (6) as a function of $\boldsymbol{\theta}$. Equation (7) then yields the human model and optimal robot commands

$$\mathbf{u}_{h,\theta}^\star(\tau) \triangleq -\mathbf{R}_{h,\theta}^{-1}\mathbf{B}(\tau)^\top[\boldsymbol{\alpha}_{h,\theta}(\tau) + \mathbf{P}_{h,\theta}(\tau)\boldsymbol{\xi}_\theta^\star(\tau)],$$
$$\mathbf{u}_{r,\theta}^\star(\tau) \triangleq -\mathbf{R}_r^{-1}\mathbf{B}(\tau)^\top[\boldsymbol{\alpha}_{r,\theta}(\tau) + \mathbf{P}_{r,\theta}(\tau)\boldsymbol{\xi}_\theta^\star(\tau)] \tag{8}$$

in a receding horizon scheme for all $\tau \in [t_c - \Delta_e, t_c]$, where $\boldsymbol{\xi}_\theta^\star$ is the optimal state deviation computed by the integration of eq. (1) with these commands, starting from $\boldsymbol{\xi}_\theta^\star(t_c - \Delta_e) = \tilde{\boldsymbol{\xi}}(t_c - \Delta_e)$, the measured state deviation at the beginning of the estimation window.

The *upper-level problem* is to find the vector $\boldsymbol{\theta}$ that best accounts for the measured interaction data on the estimation window as illustrated in Fig. 1B. To do so, the optimal solution from the lower level $[\mathbf{u}_{h,\theta}^\star(\tau), \boldsymbol{\xi}_\theta^\star(\tau)]$ is

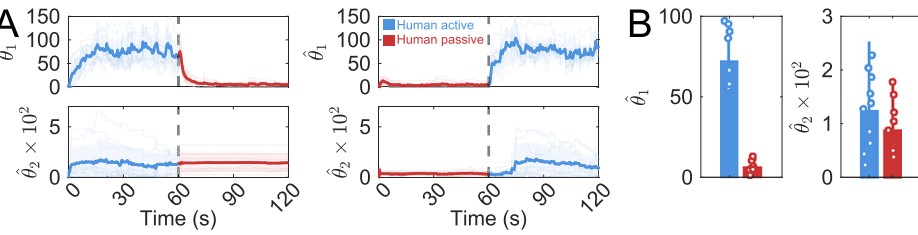

**Fig. 3 | Validation of the human cost function estimation method. A** Estimated human cost parameters $\hat{\theta}_1$ and $\hat{\theta}_2$ across all participants as a function of time when they were active in the first or second half of the trial. Individual participant values are represented by lighter-colored lines. **B** Mean values of $\hat{\theta}_1$ and $\hat{\theta}_2$ in the active and passive conditions.

compared with the actual $[\widetilde{\mathbf{u}}_h(\tau), \widetilde{\boldsymbol{\xi}}(\tau)]$ estimated from sensors measurements placed on the robot or human user during $\tau \in [t_c - \Delta_e, t_c]$. The upper-level problem can thus be formulated as:

$$\hat{\boldsymbol{\theta}} = \arg\min_{\boldsymbol{\theta}\in\Omega}\zeta(\boldsymbol{\theta}), \; \zeta(\boldsymbol{\theta}) = \int_{t_c-\Delta_e}^{t_c} [\delta\mathbf{u}_{h,\boldsymbol{\theta}}(\tau)^\top \mathbf{W_u}\, \delta\mathbf{u}_{h,\boldsymbol{\theta}}(\tau) + \delta\boldsymbol{\xi}_{\boldsymbol{\theta}}(\tau)^\top \mathbf{W_\xi}\, \delta\boldsymbol{\xi}_{\boldsymbol{\theta}}(\tau)]d\tau,$$

(9)

$$\delta\mathbf{u}_{h,\boldsymbol{\theta}}(\tau) \triangleq \widetilde{\mathbf{u}}_h(\tau) - \mathbf{u}_{h,\boldsymbol{\theta}}^\star(\tau), \; \delta\boldsymbol{\xi}_{\boldsymbol{\theta}}(\tau) \triangleq \widetilde{\boldsymbol{\xi}}(\tau) - \boldsymbol{\xi}_{\boldsymbol{\theta}}^\star(\tau), \; \Omega = [\boldsymbol{\theta}_{\min}, \boldsymbol{\theta}_{\max}],$$

where $\mathbf{W_u}, \mathbf{W_\xi}$ are positive semi-definite matrices weighting the control and state errors. Note that computing $\delta\mathbf{u}_{h,\boldsymbol{\theta}}$ requires solving coupled ordinary differential equations (see eq. (6)) making convexity guarantee of the problem difficult. For practical efficiency, we employ a local optimization algorithm presented in the Methods section.

**MPG implementation.** With $\hat{\boldsymbol{\theta}}$, the MPG controller can be implemented by solving the lower-level problem once more on the window $[t_c, t_c + \Delta_p]$. The resulting robot motor command $\mathbf{u}_{r,\hat{\theta}}^\star(t_c)$, computed from eq. (8) yields a feedback policy that allows it to correct for task errors $\boldsymbol{\xi}$ if needed. The algorithm summarizing our approach is:

**Algorithm 1**. MPG controller (with sampling time $dt$)
**Input:** Initial $\hat{\boldsymbol{\theta}} \in \Omega$
**Output:** Robot controller
  1: $t_c = 0$
  2: Estimate $\widetilde{\boldsymbol{\xi}}(t_c)$
  3: **if** $t_c \geq \Delta_e$ **do**
  4:     Update $\hat{\boldsymbol{\theta}}$ by solving the bi-level optimization eq. (9) on the estimation window $[t_c - \Delta_e, t_c]$
  5: **end if**
  6: Compute $\mathbf{u}_{r,\hat{\theta}}^\star(t_c)$ from eq. (8) using planning horizon $[t_c, t_c + \Delta_p]$
  7: Apply the robot control, $t_c = t_c + dt$ and go to step 2.

## Experiments
The proposed MPG framework can, in principle, enable a robot to achieve optimal interaction with its human user and provide appropriate assistance. However, its functioning needs to be validated. Moreover, the reciprocal adaptation between the human and robot—i.e., their co-adaptation—is unknown. Then, can a robot equipped with MPG consistently adapt to the unique characteristics of individual human operators? Conversely, will humans understand the robot's behavior and adapt their own behavior accordingly? To address these questions, we conducted three targeted experiments involving young, right-handed adults without known sensorimotor impairments. Furthermore, whether it be for the validation of the method or to study human involvement and adaptation, we will use the identified human position error weight $\hat{\theta}_1$. First, this parameter can be used to verify the validity of the method by measuring human effort and performance and analyzing whether they are consistent with the identified value. For instance, if $\widetilde{\mathbf{u}}_h$ is null, i.e., the human is passive, then the identified $\hat{\theta}_1$ should be close to zero, which we test in Experiment 1. Second, once the consistency of $\hat{\theta}_1$ with the human behavior is established, it can be used as a proxy for human involvement and performance in the task as in our

Experiments 2 and 3, a high $\hat{\theta}_1$ corresponding to a user accurately tracking the desired trajectory.

In these experiments, participants tracked a pseudo-randomly moving target displayed on a monitor positioned in front of them, using downward and upward flexion/extension movements of their right wrist to control a cursor. Their wrist was physically connected to a robotic exoskeleton implementing MPG and carrying a 1 kg load (see Fig. 2B, C). The human effort cost was normalized to 1, while position and velocity tracking weights in $\mathbf{Q}_h$ were identified during movement. The assistance parameter was set to $\lambda = 0.5$ in experiments 1 and 2 and was varied in experiment 3.

**Algorithm validation and apparent Nash equilibrium.** In the first experiment, ten participants performed the tracking task, changing between an active and a passive phases within a trial based on instructions displayed on the monitor. Each participant completed two trials: first trial starting with a 60 s long active phase followed by a 60 s long passive phase, and second trial starting with the passive phase.

As shown in Fig. 3A, the MPG resulted in stable adapted strategies, which we refer to as "apparent Nash equilibria" throughout the paper to reflect that it corresponds to the human model of eq. (2). For all participants, the identified position weight $\hat{\theta}_1$ increased significantly during the active phase to reach a plateau within approximately 15 s from the initial values $\hat{\boldsymbol{\theta}}(t_c = 0)=(5, 0.005)$. During the passive phase, $\hat{\theta}_1$ dropped rapidly to small values and remained low for the rest of the trial. A similar behavior was observed when the passive phase preceded the active phase.

This behavior was consistent across participants, as shown in Fig. 3B where the averaged values across trials of $\hat{\theta}_1$ are represented for the active and passive phase. The parameter $\hat{\theta}_1$ had higher values in the active phase compared to the passive phase ($p < 0.001$, $r > 0.5$). In contrast, the identified $\hat{\theta}_2$ values were generally too small to show a significant difference between the active and passive conditions with medium effect size ($p = 0.312$, $r = 0.267$). Therefore, for the remainder of this paper, we focus on $\hat{\theta}_1$ to analyze the consistency and tuning of the apparent Nash equilibria.

Overall, these results validate the human control model used and its parameters' identification with bi-level optimization, and show that the apparent Nash equilibrium encodes well the instructed participants' contribution to the task. However, as humans generally tend to minimize effort during learning[19,21,30], it is unclear whether they would spontaneously engage in an active interaction with the robot, or progressively rely on it with no explicit instructions.

**Consistency of the apparent Nash equilibrium.** A second experiment was conducted to investigate the emergence of apparent Nash equilibria and their inter- and intra-individual consistency. Twenty participants completed ten trials alternating 120 s long trials with MPG assistance and 60 s long trials without assistance (NA), starting with a NA trial. Figure 4A illustrates the identification results on $\hat{\theta}_1$ in the five MPG trials of a representative participant. Starting from the initial value, the human cost estimates consistently increased and stabilized in about 20 s.

Figure 4B displays the plateau values across participants, based on a convergence criterion (see Methods). Considering the detected plateau values of $\hat{\theta}_1$ over all MPG trials, a decomposition of variance indicated that 26% of the total variability was related to variability within participants and about 74% to differences between participants. This indicates subject-

**Fig. 4 | Inter- and intra-individual consistency of the apparent Nash equilibrium. A, B, C, H** display the results over consecutive MPG trials, while **D, E, F, G** presents results for the first and the last trials without robot assistance (NA) in beige color and with robot assistance (MPG) in blue. **A** Evolution of $\hat{\theta}_1$ as a function of time with MPG assistance for a representative participant. Detected plateaus (**B**) and applied torque (**C**) across all trials with MPG for each participant. RMS of the human applied torque (**D**) and of normalized EMG signals from a wrist antagonist muscle pair (**E**). RMS of wrist angular position error (**F**) and velocity error (**G**) for all participants. **H** Improvement of average coordination index (CI) across the whole population for successive trials. The solid line represents the population average and the shaded area standard deviation across the population.

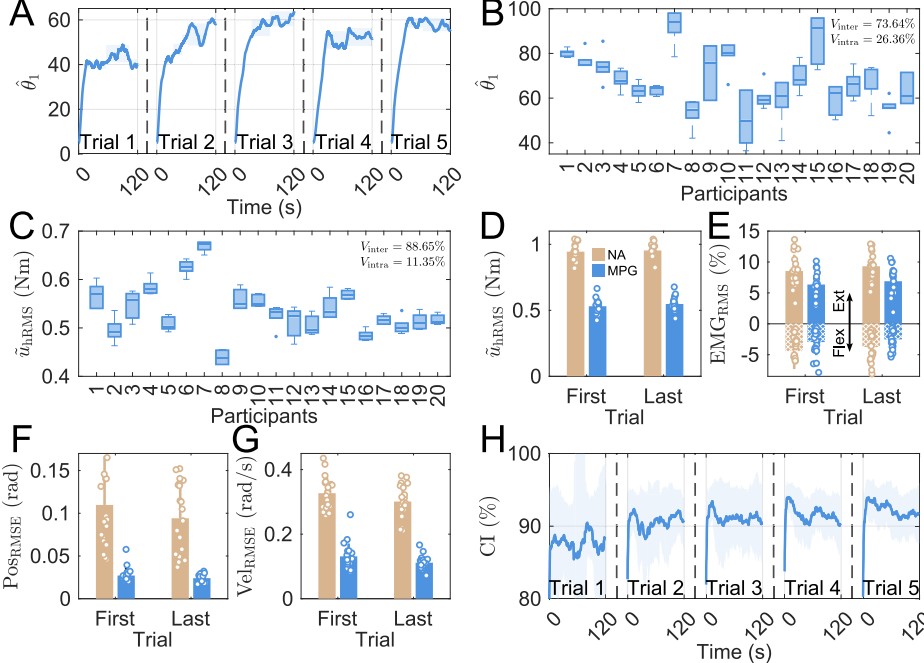

specific Nash equilibria, resulting in differing effort contributions across participants (Fig. 4C). However, the correlation of $R = 0.686$ ($p < 0.001$) between $\hat{\theta}_1$ and $\tilde{u}_{hRMS}$ shows that these quantities are not strictly equivalent, likely due to anthropometric and volitional control differences.

The MPG controller reduced significantly the human effort during the task. As shown in Fig. 4D, the mean human torque $\tilde{u}_{hRMS}$ was higher in the NA condition compared to MPG ($p < 0.001$, $r > 0.5$). Similarly, muscle activity RMS was reduced in both the extensor ($p < 0.001$, $r > 0.93$) and flexor ($p < 0.001$, $r > 0.5$) muscles, even though there is little flexor activity as gravity is facilitating downward motion (Fig. 4E).

MPG assistance clearly improved task performance (Fig. 4F,G), as it significantly reduced the position and velocity tracking errors relative to NA ($p < 0.001$ with $r > 0.5$). On the other hand, there were no significant differences in tracking performance between the first and last trials, indicating consistent performance with MPG from the first trial onward.

Figure 4H illustrates that the CI, defined as the correlation coefficient between human and robot torques over a 10 s sliding window, increased across trials and participants (comparison of first versus last trial: $p < 0.01$, $r > 0.5$), while the inter-individual variability of the CI tended to decrease. These findings suggest that all the participants learned to better understand the robot, enabling the robot to more effectively identify and respond to their control dynamics. Additional results are provided in Supplementary Results.

These results illustrate effective co-adaptation between the participant and the robot, as they learn to coordinate their actions over time. Interaction with MPG leads to stable, subject-specific apparent Nash equilibria yielding efficient assistance.

**Modulating the apparent Nash equilibrium.** The third experiment assessed if and how the assistance meta-parameter $\lambda$ influences the human interaction behavior. The same 20 participants as in the second experiment completed eight 120 s long trials with different values of $\lambda \in \{0, 0.5, 0.7, 1\}$. Each value was used for two trials presented in a random sequence.

Figure 5A-C illustrate the influence of $\lambda$ on the induced behavior for a representative participant. We see that the identified $\hat{\theta}_1$ was influenced by the value of the assistance parameter $\lambda$ in a monotonic manner, suggesting a systematic shift of the apparent Nash equilibrium. A larger $\lambda$ increased the

robot assistance, effectively lowering the human's contribution to the task in torque (Fig. 5B) and EMG (Fig. 5C).

These trends are confirmed in the population-level results of Fig. 5D–F. The identified $\hat{\theta}_1$ changes with $\lambda$ (Fig. 5D) ($p < 0.001$, $W = 0.68$, $\chi_3^2 = 40.92$), decreasing with increasing $\lambda$ (post-hoc tests with $p < 0.012$ except for $\lambda = 0$ and $\lambda = 0.5$). This result shows that the participant's apparent Nash equilibrium can be shifted through the robot's assistance. The contributions of the human and robot torque change inversely with larger $\lambda$ ($p < 0.001$, $W = 0.66$, $\chi_3^2 = 79.38$) for the human torque with post-hoc tests showing $p < 0.0012$ for all $\lambda$ values; $p < 0.001$, $W = 0.69$, $\chi_3^2 = 83.67$ for the robot torque with post-hoc tests showing $p < 0.001$ for all $\lambda$ values, lowering the human effort and increasing the robot effort (Fig. 5E). The effort also decreases in the extensor muscle ($p < 0.001$ with a high effect size $W = 0.58$, $\chi_3^2 = 70.14$) with increasing $\lambda$ with post-hoc tests showing $p < 0.0013$ for all $\lambda$ values (Fig. 5F). Importantly, beyond this reduction of extensors' activity, we did not observe any change in co-contraction– that is a simultaneous activation of both flexors and extensors resulting in increased joint stiffness without generating movement. This confirms that the decreased net joint torque obtained through inverse dynamics did not mask an increased co-contraction effort.

We then analyzed whether the apparent Nash equilibrium exhibited by individuals was robust across different values of $\lambda$, which would show a consistency of the user-specific assistance provided by MPG, even for different targeted human and robot contributions. To test this premise, we computed the average $\hat{\theta}_1$ across the population for each $\lambda$, and the distance to this population average for each participant and each $\lambda$. Then, we conducted Pearson correlation tests to analyze whether an individual's distance to the population average was robust across conditions. These tests showed that the identified individual strategies were consistently ranked across participants and conditions for $\lambda < 1$ ($\lambda = 0$ vs 0.5 and 0.7: $r > 0.5$, $p < 0.015$). However, the participants' behavior with $\lambda = 1$ were not correlated to their behavior in the other conditions (all comparisons: $r < 0.11$, $p = 0.33$). With the same analysis strategy, a similar pattern was observed in participants' effort contributions (i.e., average torque): conditions with $\lambda \neq 1$ were correlated (all comparisons: $r > 0.5$, $p < 0.015$), while the condition $\lambda = 1$ was not correlated to other conditions (all comparisons: $r < 0$, $p = 0.08$). These results suggest that $\lambda < 1$ gives more flexibility to participants, allowing them to

**Fig. 5 | Effect of $\lambda$ on performance, effort and human-robot interaction.** How the position weight parameter $\hat{\theta}_1$ (**A**), wrist torque (**B**) and normalized RMS of the wrist extensor muscle (**C**) depend on $\lambda$ for a representative participant. Dependency on $\lambda$ of the mean position weight parameter $\hat{\theta}_1$ (**D**), human and robot applied torque effort (**E**) (with solid lines representing the average and shaded lines individual values), RMS of an antagonist wrist muscle pair (**F**), joint angle position (**G**) and velocity (**H**) error, and coordination index (**I**) across all participants.

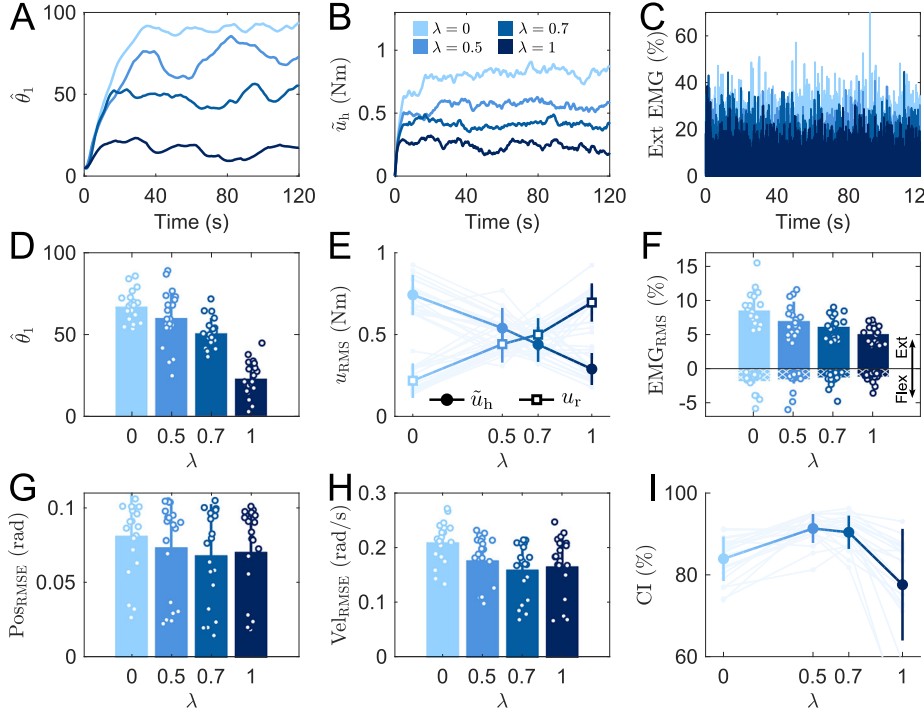

express user-specific differences. Conversely, when $\lambda = 1$, the robot's contribution to the task is largely aimed at reducing human inputs, resulting in an homogenization of behaviors as the robot takes the lead while the human becomes more passive.

However, the tracking performance remains consistently strong with different $\lambda$ values (Fig. 5G, H). The parameter $\lambda$ had no influence on position error ($p = 0.048$, Friedman test, with no significant pairwise difference after correction) and slightly decreased the velocity error ($p < 0.001$, $W = 0.17$, $\chi^2_3 = 20.46$, with pairwise significant difference only between $\lambda = 0$ and $\lambda = \{0.7, 1\}$, $p < 0.05$). Note that in this test the statistical power was 0.60. For all the other reported tests of the present paper, the statistical power was >0.8.

Finally, we investigated the effect of $\lambda$ on the CI, i.e., the correlation between the human and robot torques (Fig. 5I). $\lambda$ has a significant effect on CI ($p < 0.001$, $W = 0.67$, $\chi^2_3 = 40.44$, Friedman). $\lambda = 0.5$ and $\lambda = 0.7$ lead to higher CI than $\lambda = 0$ and $\lambda = 1$ ($p < 0.001$, $r > 0.1$). These results indicate that an optimal coordination is achieved around $\lambda = 0.5$, potentially reflecting a better balance in the sharing of task control, where each agent carries about half of the load (Fig. 5E).

These results demonstrate that the human interaction behavior is strongly influenced by the assistance parameter $\lambda$. The human contribution to the task is modulated by $\lambda$, providing a means to guide participants toward a desired interaction behavior.

## Discussion

The control of contact robots has been extensively studied over the past two decades, particularly to facilitate neurorehabilitation and collaborative work in manufacturing[3-5,11]. For such applications, robots are required to interact with users while optimally assisting them, assessing their condition, and inducing behaviors to improve physical training (Fig. 1). However, existing interaction control methods do not fulfill all these requirements. Most approaches adapt control based on the performance of the human-robot system without explicitly modeling the human user. This includes heuristic methods[31], iterative and adaptive control methods using gradient-descent minimization of feedback error to identify a basic environment model[3,6,7,32,33], and human-in-the-loop optimization of the user's energy expenditure[9,10]. As these methods are not designed to

consider the human user's motor plan, they cannot provide proactive assistance during movement. Model predictive control offers a solution to this issue by allowing the robot controller to optimally exploit the upcoming human control over a short time horizon[34]. Leveraging muscle activity measured from EMGs and the electromechanical delay[34], developed an assistance method that optimally adjusts to the forthcoming human control. However, transmission latencies and filtering issues inherent to EMG processing drastically limit the usable time horizon[35], rendering such methods impractical for providing a truly proactive assistance. Furthermore, the in-field implementation of EMG-based assistive controllers is also hampered by practical limitations, such as the complex and time consuming process of installing the sensors and calibrating the EMG-to-torque model[36]. In addition, EMG signals are noisy and require prior filtering, which is time consuming in real time control scenarios. Finally, the quality of EMG signals tends to decrease through time due to environmental conditions, such as sweat between the skin and the electrodes. Yet, EMG could be useful to estimate physiological parameters such as fatigue, which could in turn provide relevant information to modulate the $\lambda$ parameter in MPG.

These issues can be addressed by adopting a principled approach to model the upcoming human control, as proposed in Ref. 12, which considers the human and robot as two agents and leverages differential game (DG) theory to generate appropriate motor commands. Existing DG algorithms for contact robots[13-15] assume an infinite planning horizon for the human agent, whereas humans naturally plan their movements over a finite horizon[16]. For its application, this setting requires a time-invariant system with null steady-state effort (as detailed at p.337 in Ref. 18), conditions that are not necessarily satisfied in trajectory tracking or load-lifting assistance tasks. Therefore, implementing an infinite-horizon DG in our context necessitates several ad hoc assumptions, resulting in coarser approximations and reduced shared control abilities (see Supplementary Note 2). Moreover, previous infinite-horizon formulations can lead to the paradox that the human cost cannot be identified when the task is well performed and no error is observed[13,14]. All these limitations are naturally resolved by the finite-horizon MPG controller introduced in this paper, as demonstrated by our experiments and the simulations presented in the Supplementary Note 2.

While previous DG-based control studies[13–15,17] only tested the functioning of the developed algorithms, our experiments provide an in-depth analysis of the co-adaptation between human and robot:

- The first experiment demonstrated the effectiveness of the MPG algorithm for real-time identification of human model parameters, and optimal interaction exploiting the upcoming human control dynamics.
- The second experiment examined *how the robot adapts to human users*. It showed that MPG identifies individual cost functions in a stable and temporally consistent manner.
- The third experiment examined *how humans adapt their behavior in interaction with the robot*, where we found that their behavior and effort were finely tuned by the assistance meta-parameter of the robot cost function.

The successful human-robot co-adaptation with MPG was also illustrated by the increasing coordination of their movements. Earlier motor control studies showed that humans tend to minimize their effort with practice[19,21]. However, the participants to the third experiment did not maximally rely on the robot. Conversely they engaged in the task by actively collaborating with it. Finally, note that the individual cost functions may serve as a biomarker for human motor control, and to develop tailored physical training and rehabilitation protocols guided by the assistance meta-parameter. Importantly, we have shown that this biomarker can be identified for behaviors ranging from passivity to full engagement in the task performance. However, the present study does not address what would happen in impaired individuals, for instance in post-stroke patients with spasticity that could result in more complex interactions. This important limit could be addressed in simulation studies, e.g., showing that increased motor noise or viscoelasticity does not prevent the method from identifying the human costs[37], and in clinical studies with impaired patients.

Do humans interacting with a robot behave as predicted by differential game theory? Although[38,39] provided some evidence that human motion aligns with a Nash equilibrium after learning, the online control strategy employed by humans may significantly differ from the MPG algorithm which is used as a tool to control the robot. In fact, the average RMSE between the measured and predicted participants' torque was ≈0.3 Nm (see Supplementary Results), which was mostly due to an under-estimation of the human control inputs. This under-estimation likely results from the under-estimation of errors induced by the human as a consequence of sensorimotor noises, as they are not accounted for by the human model. To overcome this limit, the human model may need to be formulated as a stochastic optimal control problem so as to account for this noise[40] and external uncertainty[41], e.g., related to the robot. Nevertheless, the fact that the participants skillfully adapted their behavior to the robot's commands–whether it be through learning in the second experiment or through different robot contributions in the third experiment–indicates that they understood the robot strategy and considered it to control their movement. In this sense, *our results offer compelling evidence that the human interactive behavior is controlled in game theory like manner*.

The concept paper[12] used a game theoretical framework to identify and characterize representative human-robot interaction behaviors. With maximal *assistance* the robot dedicates as much energy as needed for the task so as to minimize human effort. *Collaboration* attempts to share effort equally with the human, *education* to promote efficient learning of a physical task by bringing the human to proactively contribute to it, and *competition* to challenge and strengthen the human. Dedicated DG algorithms have been developed for some of these behaviors:[14] implemented a collaborative behavior, and[13,15] a competitive assist-as-needed learning strategy. Our third experiment demonstrated that by enabling dynamic adjustment of the robot's assistance level, the $\lambda$ parameter induces a continuum of interaction behaviors from maximal to minimal *assistance* over equal *collaboration*. *Education* can be implemented by modulating the assistance level —for instance, starting with maximum assistance to guide the human's

movement and gradually reducing it to encourage the human to become increasingly proactive. Finally, *competition* can be implemented by incorporating a zero-sum game condition as was done in Refs.13,15. The simplicity and versatility of the MPG framework make it a powerful tool for inducing diverse interaction behaviors while reasoning only in terms of cost functions, with significant potential for developing efficient physical training and rehabilitation systems. This large panel of possible interaction scenarios further allows the MPG framework to be more flexible in setting the robot's assistive contribution than recent alternatives, for instance based on reinforcement learning[42] or on adaptive impedance control[7]. Interestingly, future works could focus on combining learning-based approaches and MPG, in particular by using reinforcement learning for the estimation of the human cost function. Such approaches could be very powerful to mitigate the curse of dimensionality when solving the inverse problem in a multi-joints setting.

The current implementation of MPG requires prior knowledge of the intended motion control for some time horizon to determine a local approximation. This intended motion could be obtained by estimation methods such as in Refs. 27,28, or by formulating interaction control as a nonlinear and non-quadratic DG problem[43] in a one stage process. Such extended MPG framework may predict how, during interaction, the agent with more accurate sensorimotor information increases their impedance control gain and leads the action[44,45]. Furthermore, the present methodology relies on the assumption that control inputs and state variables are unbounded. Additional work would be needed to relax this assumption by considering methods that allow control/state constraints in the direct DG solving and that possibly seek to identify both the human cost function and control boundaries, which could be important with pathological participants (e.g.,[46]). Finally, while we have shown in a recent simulation study that the present method is scalable to multi-joint cases in principle[37], future works will need to investigate whether humans can adapt to MPG-controlled robots in multi-joint tasks, where human-robot coordination may be more difficult to learn. However, the one-DOF system validated through this paper's experiments is already suitable for neurorehabilitation applications[47], where training joint-by-joint appears to be as effective as with multi-joint movements[48].

## Methods

In this section, we first provide details regarding the experimental setup and task used for the three experiments, followed by the resulting human-robot interaction dynamics. Then, we detail the implementation of the MPG controller and of the bi-level identification for our specific interaction dynamics. Finally, we provide the anthropometric data of the tested participants and details of the protocol for each of the three experiment, followed by the employed data processing techniques and statistical tests.

### Experimental setup and task

The protocols for the three experiments were approved by the ethical committee for research (CER-Paris-Saclay-2022-071), and written informed consent was obtained from each participant prior to performing an experiment.

The task consisted in tracking a smoothly but pseudo-randomly moving target displayed on a monitor using wrist flexion/extension movements in the vertical plane (Fig. 2C). Participants were connected to an HRX-1 robotic wrist exoskeleton (HumanRobotiX, UK) controlled at 100 Hz, which carried a 1 kg load to increase muscle effort and enhance the effects of robotic assistance (Fig. 2B). Two EMG electrodes (Cometa MiniWave, Italy) were placed on the flexor carpi radialis and extensor carpi ulnaris (following SENIAM guidelines[49]) to measure the activity of a wrist antagonist muscle pair.

Visual feedback of the desired trajectory and participant's current wrist angle were provided on a monitor placed in front of them. The current wrist angle was represented by a 2 cm diameter disk (Fig. 2C). This cursor's horizontal position on the monitor was fixed, while its vertical position was

computed as an affine mapping of the wrist angle. The trajectory to track was moving towards the cursor at a velocity of 34.92 cm/s, with a length displayed on the screen of 52.38 cm.

## Interaction dynamics

The dynamics of the robot (including its attached load) and the human wrist were modeled respectively as:

$$\begin{cases} I_r \ddot{q}_r = u_r - m_r g \ell_r \cos(q_r) - \mathcal{D}\dot{q}_r - \gamma \\ I_h \ddot{q}_h = u_h - m_h g \ell_h \cos(q_h) + \gamma \end{cases} \quad (10)$$

where $\gamma$ is the interaction torque between the robot and human, $I_r$, $I_h$ are their respective moments of inertia relative to the wrist axis, $m_r$, $m_h$ their masses, $\ell_r$, $\ell_h$ their lengths to the center of mass, and $\mathcal{D}$ is the robot damping coefficient. The robot parameters were identified in a separate procedure before the experiments ($I_r = 0.01 \, \text{kg} \cdot \text{m}^2$, $\mathcal{D} = 0.006 \, \text{kg} \cdot \text{m}^2 \cdot s^{-1}$, $\ell_r = 0.09 \, \text{m}$, $m_r = 1.2 \, \text{kg}$), while the participants' parameters were estimated from anthropometric tables[50].

Assuming that the wrist flexion/extension and robot joints are aligned, thus $q \equiv q_h \equiv q_r$, the coupled {human+robot+load} system was modeled as:

$$\mathbf{f}(\mathbf{x}, u, t) = \begin{bmatrix} \dot{q} \\ (u_h + u_r - \mathcal{G}(q) - \mathcal{D}\dot{q})/\mathcal{I} \end{bmatrix} \quad (11)$$

where $\mathbf{x} = (q, \dot{q})$ is the state vector with joint position and velocity, $u = u_r + u_h$ is the total input torque with $u_r$ the robot torque and $u_h$ the human torque, $\mathcal{I} = I_h + I_r$ the moment of inertia of the system relative to the rotation axis, and $\mathcal{G}(q) = \mathcal{K} \cos(q)$ the gravitational term with $\mathcal{K} = (m_h \ell_h + m_r \ell_r)g$, $g = 9.81 \, \text{m.s}^{-2}$. These dynamics are nonlinear due to the sine function, and were linearized around a desired trajectory-control pair $(q_d, u_d)$ as:

$$\dot{\boldsymbol{\xi}} = \begin{bmatrix} 0 & 1 \\ \mathcal{K}\sin(q_d)/\mathcal{I} & -\mathcal{D}/\mathcal{I} \end{bmatrix} \boldsymbol{\xi} + \begin{bmatrix} 0 \\ 1/\mathcal{I} \end{bmatrix}(u_h + u_r)$$
$$- \begin{bmatrix} 0 \\ (\mathcal{I}\ddot{q}_d + \mathcal{G}(q_d) + \mathcal{D}\dot{q}_d)/\mathcal{I} \end{bmatrix}, \boldsymbol{\xi} = \begin{bmatrix} q - q_d \\ \dot{q} - \dot{q}_d \end{bmatrix}, \quad (12)$$

where all variables depend on time.

## MPG controller implementation

No final cost was used in the experiments, i.e., $\mathbf{Q}_h^f = \mathbf{Q}_r^f = \mathbf{0}$. The human model cost function eq. (2) was parametrized with $\mathbf{Q}_{h,\theta} = \text{diag}(\theta_1, \theta_2)$ and $R_h = 1$, yielding $J_h(\boldsymbol{\theta})$. According to equations (3)–(4), the cost function of the robotic wrist exoskeleton was:

$$J_r(\lambda) = \frac{1}{2} \int_\tau^{\tau_f} \left[ \boldsymbol{\xi}(t)^\top \mathbf{Q}_r \boldsymbol{\xi}(t) + 0.5 \, u_r^2(t) + 0.5 \, u_h^2(t) + (1-\lambda) \, u_r^2(t) + \lambda \, u_h^2(t) \right] dt .$$
$$(13)$$

The angle from the HRX-1 encoders signal was smoothed using a moving average filter with a 5-sample window size before numerical differentiation, enabling real-time computation of velocity and acceleration. These signals were used to estimate the human wrist torque $u_h$ during movement based on the dynamics in eq. (12). The parameters of the bi-level optimization were set as: $\Delta_p = 1.5 \, \text{s}$, $\Delta_e = 0.25 \, \text{s}$, $\hat{\boldsymbol{\theta}}(t_c = 0) = (5, 0.005)$, $W_u = 1$, $\mathbf{W}_\xi = \mathbf{0}$, with boundaries $\hat{\boldsymbol{\theta}}_{min} = (0, 0)$ and $\hat{\boldsymbol{\theta}}_{max} = (+\infty, 0.1)$. The robot cost was $\mathbf{Q}_r = \text{diag}(30, 0.1)$ for all experiments and all participants, which was below the stability boundaries evaluated in simulation as $\text{diag}(25500, 25)$. In addition, the applied robot torque, first computed through Eq. (7), was then saturated at 2 Nm for safety reasons and to comply with hardware constraints. The assistance parameter was set to $\lambda = 0.5$ in experiments 1 and 2. It was varied between 0 and 1 in experiment 3 as described in the Results.

## Real-time bi-level optimization

To compute the MPG control online, the finite-horizon DG problem was solved for $\hat{\boldsymbol{\theta}}$. For this dyadic affine-quadratic case, the closed-form solutions allowed computations at the rate of the robot control loop. If needed, the feedback form of the robot control policy could enable to apply the controller for some time $< \Delta_p$ without re-computation, though this was not tested here.

Estimating the human cost parameters $\hat{\boldsymbol{\theta}}$ online is more challenging due to the underlying bi-level optimization. The receding horizon scheme further complicates matters because computing $\mathbf{u}_{h,\theta}^\star(\tau)$ requires solving several DG problems, with the number growing in proportion to the length of the estimation window ($\Delta_e$) and the control rate. However, waiting for convergence of the bi-level optimization may not be necessary or desirable because: (i) observations are corrupted by noise, so refining the solution $\hat{\boldsymbol{\theta}}$ until convergence could lead to overfitting; (ii) if new observations become available during the optimization process, it may be better to exit the optimization loop before full convergence and initialize the next optimization with the most recent estimate $\hat{\boldsymbol{\theta}}$. Since the previous interaction data $[\tilde{\mathbf{u}}_h(t), \tilde{\boldsymbol{\xi}}(t)]$ for $t \in [t_c - \Delta_e, t_c]$ may evolve only slightly between successive instances of the bi-level optimization, the update rate of $\hat{\boldsymbol{\theta}}$ could be slower than the robot's control rate. Therefore, a maximum number of iterations for the upper-level optimization can be adjusted depending on the problem's dimensionality and control rate. Finally, note that the planning and estimation problems are independent and can be handled in parallel.

Here a derivative-free optimization method was employed to solve the upper-level optimization problem and improve the estimates of $\boldsymbol{\theta}$ throughout task execution. We used Powell's Bound Optimization BY Quadratic Approximation method (BOBYQA), that employs an iterative quadratic approximation of the objective function in Eq. (9), then minimizes it using the conjugate gradient procedure[51]. It is a local, derivative-free optimization method that proved to be suited and robust for our purpose in preliminary simulation studies. In simulation works, we have shown that our method can robustly identify correct human costs during 1-DoF movements with respect to different initial guesses[17] and with respect to noise/modeling errors in 2-DoF movements[37]. Other derivative-free optimization methods could be used if needed (see[52] for a review), as well as gradient-based methods leveraging algorithmic differentiation for differential equation solvers. In our context, gradient-based algorithms led to less stable convergence of the human costs estimation, which is why gradient-free ones were privileged. The BOBYQA algorithm has several parameters. The optimization procedure was initialized at each control step with the most recent $\hat{\boldsymbol{\theta}}$ available. Lower bounds $\boldsymbol{\theta}_{min} = [0, 0]$ and upper bound $\boldsymbol{\theta}_{max} = [200, 0.2]$ were set to ensure numerical stability based on simulations. The default initial trust region radius was set to $\boldsymbol{\theta}_{max}/2000$. When the error $\zeta$ in eq. (9) fell below a threshold of 0.25, corresponding to an absolute mean error of 0.1 Nm in terms of torque, the initial trust region radius was updated by multiplying its default value by $\zeta$. The maximum of $\zeta$ function evaluations was fixed to 6 to ensure that an updated $\hat{\boldsymbol{\theta}}$ could be obtained in less than 10 ms, in line with our robot control rate. The implementation of BOBYQA provided by the Matlab NLopt toolbox was used[53]. Note that the computational time can be further reduced by leveraging parallel computing techniques.

## Experiment 1

The first experiment was designed to test the inverse DG method with human participants, instructing them to remain passive or active in the first or second half of a trial. This experiment involved 10 participants (5 females) with age $= 23.5 \pm 4.5$ years, height $= 1.70 \pm 0.09$ m, and weight $= 61.3 \pm 6.99$ kg. The target trajectory was defined (in radians) as:

$$q_d(t) = \frac{\pi}{16}[1 - \cos(0.97t) - \cos(2.34t) - \cos(4.11t)]. \quad (14)$$

As this trajectory's period is large relative to the task duration, it appeared as pseudo-random to the participants. Half of the participants were asked to

actively track the displayed trajectory for the first 60 s and remain passive for the second half of the trial, and conversely in a second trial. The other participants performed the two trials in reversed order. No trajectory was displayed during the passive period while the monitor instructed participants to "relax". Note that participants were given breaks of one-minute between trials in all experiments to prevent fatigue.

## Experiments 2 and 3

Another 20 participants (with age = 23.05 ± 3.73 years, height = 1.76 ± 0.08 m, and weight = 73.80 ± 13.36 kg, 3 females) carried out the second and third experiments, designed to assess their co-adaptation with the MPG controlled robot. The desired trajectory was generated using third order B-splines, allowing randomization both within and between trials to minimize the learning of trajectory patterns. The 240 B-splines control points were uniformly distributed on the interval [0, 0.9] rad with one every 0.5 s, generated using Matlab's rand function. The trajectory was displayed on a sliding finite horizon $\Delta_p = 1.5$ s.

In experiment 2, each participant performed a series of ten trials, alternating 120 s long trials with MPG assistance with $\lambda = 0.5$, and 60 s long trials without assistance (NA), starting with a NA trial.

In experiment 3, each participant performed eight 120 s long trials with different values of $\lambda \in \{0, 0.5, 0.7, 1\}$. Each value was used in two trials, in a random sequence.

## Data analysis

After the experiment, $\widetilde{u}_h$ was low-pass filtered for analysis using a second-order Butterworth filter with 5 Hz cut-off frequency. EMG signals were band-pass filtered for analysis using a fourth order Butterworth filter with cut-off frequencies [20, 450] Hz[54].

The stability of the estimated cost weight $\hat{\theta}_1$ was quantified by identifying plateaus in individual trials. A plateau was defined as a time window where the difference between the maximum and the minimum values of $\hat{\theta}_1$ remain below 5% of $\theta_{max,1}$ for at least 10 seconds. The average value of $\hat{\theta}_1$ on the identified plateaus was then used to assess the consistency of the estimated human cost.

We first conducted a Shapiro-Wilk test to verify whether our data was normally distributed. This test showed that our data violated the assumption of normality. Therefore, to examine the effects of experimental conditions on interaction metrics, a non-parametric Friedman test was performed with effect sizes reported as Kandall's $W$ for significant outcomes. For pairwise comparisons of conditions, non-parametric Wilcoxon signed-rank tests were conducted, and the effect sizes were reported as the rank-biserial correlation $r$. Statistical significance was set at $p < 0.05$, and adjustments for multiple comparisons were made using the Bonferroni method. Finally, we performed *a posteriori* power analyses for all the reported statistical tests using G-power, which showed that all but one tests had a high power >0.8.

The decomposition of variance used to analyze the consistency of the apparent Nash equilibria was based on the sum-square (SS) output of the Friedman test:

$$V_{inter} = \frac{SS_{inter}}{SS_{inter} + SS_{intra}} \times 100 \quad \text{and} \quad V_{intra} = \frac{SS_{intra}}{SS_{inter} + SS_{intra}} \times 100 \tag{15}$$

To test the robustness of the individual apparent Nash equilibrium, we conducted pairwise Pearson rank correlation tests on the difference between population average behaviors and individual behaviors between different conditions. We report the Person correlation coefficient $r$ and the corresponding $p$-values.

## Ethics declarations

The protocols for the experiments conducted in this study were approved by the ethical committee for research (CER-Paris-Saclay-2022-071), and written informed consent was obtained from each participant prior to performing an experiment.

## Data availability

The datasets generated and analyzed during the current study are available in the ZENODO repository, https://doi.org/10.5281/zenodo.15241994.

## Code availability

The custom code used in this study is available from the corresponding author upon reasonable request.

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

## Acknowledgements
This study was funded by the French Agence Nationale de la Recherche (EXOMAN project, grant number ANR-19-CE33-0009).

## Author contributions
Conceptualization—A.H., D.V., O.B., E.B., B.B. Methodology—A.H., B.B. Software—A.H. Validation—A.H. Formal analysis—A.H. Investigation—A.H., A.F. Resources—O.B., B.B. Data Curation—A.H. Visualization—A.H.,

D.V., E.B., B.B. Supervision—O.B., B.B. Project administration—O.B., B.B. Funding acquisition—O.B., B.B. Writing—Original Draft—A.H., A.F., D.V., E.B, B.B. Writing—Review and Editing—all authors.

## Competing interests

The authors declare no competing interests.
