## [Transparent Peer Review file · Communications Engineering]

Model predictive game control for personalized and targeted interactive assistance

Corresponding Author: Professor Bastien Berret

Version 0:

Reviewer comments:

Reviewer #2

(Remarks to the Author)

This paper introduces a model-predictive game (MPG) controller for personalized physical human-robot interaction. A finite-horizon differential game framework uses receding-horizon Nash equilibrium to model human-robot co-adaptation, explicitly accounting for human limited planning horizon. Experimental validation with 30 participants demonstrates bidirectional adaptation. The robot identifies stable, individualized human control strategies, while humans adapt to the robot. Here are the comments on this paper.

1. Differential game theory is typically applied to non-cooperative adversarial scenarios. Please explain the motivation for utilizing this theoretical tool in controller design.
2. The paper merely mentions the limitations of traditional DG algorithms but provides no quantitative comparisons against mainstream human-robotic interaction control methodologies.
3. How does the combination of inverse DG, receding horizon, and λ -based behavior modulation distinguish MPG from prior work?
4. The methodology description focuses exclusively on optimization objectives yet omits detailed specifications of optimization constraints. Critical practical implementation considerations, such as bounded control inputs, remain unaddressed.
5. EMG signals are recorded but minimally analyzed. How does EMG inform the human effort model or could refine the inverse DG optimization?
6. Does the discussion of model parameters sufficiently characterize complex human behaviors? Is it necessary to incorporate additional physiological indicators?
7. The experiment participants consist exclusively of "young, right-handed adults," without validation across diverse populations—such as individuals of varying ages or pathological conditions like motor impairments.
8. The MPG approach relies on the online solving of differential games, yet the paper fails to report computation time metrics or validate the real-time feasibility of the control algorithm.
9. Validation is limited to 1-DOF wrist control. How does the proposed method address scalability to multidimensional systems?
10. The recency of the references is insufficient. Please incorporate citations of state-of-the-art advances from the last three years.

Reviewer #3

(Remarks to the Author)

The major claims of the paper is a model predictive game (MPG) c for physical human-robot interaction (pHRI) control. The current version is suggested as major revision. Please check following comments

1) The MPG is novel and will be of interest to others in the community and the wider field. However, its novelties has not yet been highlighted by comparing state-of-the-art(SOTA) such as adaptive impedance control [1, 2] in exoskeleton experiments.

[1] Zhao, Yihui, et al. "Adaptive cooperative control strategy for a wrist exoskeleton using model-based joint impedance estimation." *IEEE/ASME Transactions on Mechatronics* 28.2 (2022): 748-757.

[2] X. Xiong et al. "Learning-based multifunctional elbow exoskeleton control." *IEEE Transactions on Industrial Electronics* 69.9 (2021): 9216-9224.

2) The conclusions and discussion are incomplete, it would be helpful if you could add references and contents on i) the advantages and extensions of the proposed MPG over neural network based control and adaptive impedance control in multi-joints [3, 4].

[3] Li, Sitan, and Chien Chern Cheah. "Deep Neural Network-Based Jacobian Control of Robot Manipulators: Offline Regression and Online Adaptation." *IEEE/ASME Transactions on Mechatronics* (2025).

[4] "Online sensorimotor learning and adaptation for inverse dynamics control." *Neural Networks* 143 (2021): 525-536.

3) the contribution of the proposed should be clearly stated by comparing SOTA [1,2,4]. Adding a comparison table should be helpful.

4) A multi-jointed wearable robot was included in Fig. 1, but the validation is based an one-DOF exoskeleton in Fig.2. It is misleading.

Reviewer #4

(Remarks to the Author)

Introduction

1. The abstract and introduction mention "assistance meta-parameter," but the mechanism and implications are not clearly intuitive until much later (or just mentioned a little in Figure 1). Consider previewing how this parameter works earlier for better reader orientation.

2. The authors state that "co-adaptation between a human and a contact robot using a DG controller has not been thoroughly investigated before." This is a strong claim. If possible, temper it by acknowledging adjacent studies or explaining what makes their investigation distinct.

Results

Interaction control as differential game on a receding finite horizon

3. Consider adding brief intuitive descriptions alongside the equations to improve accessibility for the readers unfamiliar with control theory. (For example, ξ represents deviation from the planned trajectory, which the robot attempts to minimize...)

4. It would be helpful to explain why the linearization around a planned trajectory (as in Eq. 1) is valid in practical scenarios.

5. It would help to explicitly state or remind the reader what the intuition is behind the backward-in-time integration (Eq. 6).

Inverse DG for human cost estimation

6. A brief justification of why diagonal matrices are assumed for Q_h, Q_{fh}, R_h would be useful.

7. For the cost function in Eq. (9) it may be useful to discuss whether the problem is convex or how optimization is handled numerically, especially since θ appears in a nonlinear fashion through the solutions of Eq. 6 and 8.

8. Can you discuss a little bit about the robustness or sensitivity for the inverse DG and bi-level formulation?

Experiments

9. All three experiments rely heavily on the interpretation of $\hat{\theta}_1$ as a proxy for human intent or engagement. It would strengthen the narrative to briefly justify this assumption (e.g., why $\hat{\theta}_1$ is a valid or sufficient proxy) early in the section or as a transition to Experiment 1.

10. It would be helpful to include whether assumptions of normality were tested, especially for ANOVAs or Friedman tests.

11. For Experiment 1, you might consider including effect size (r) for the non-significant results on $\hat{\theta}_2$ to contextualize their potential practical (vs. statistical) insignificance.

12. For Experiment 2, the decomposition method for variance is not described. A brief mention or reference in the Methods section would help readers replicate or verify this analysis.

13. The fact that CI increases while its variability decreases is a nice indicator of co-adaptation. Consider highlighting whether this improvement is more strongly driven by changes in the human or robot strategy.

Discussion

14. The discussion of EMG-based model predictive control ([29], [30]) could be improved by clarifying what specifically limits its practical use.

15. When discussing user engagement despite no explicit instruction (e.g., in Experiment 3), consider noting any observed variation between participants.

16. The paragraph could be improved by explaining how close human behavior actually comes to MPG predictions.

Methods

17. Consider adding a brief overview paragraph before the detailed breakdown to help the reader mentally organize the

coming subsections.

18. Please clarify whether the sample size for each experiment was determined based on a power analysis.

19. Consider adding a sentence justifying why gradient-free methods are preferred over gradient-based ones in the BOBYQA setup.

Version 1:

Reviewer comments:

Reviewer #2

(Remarks to the Author)

1. It seems that the work is quite similar with the previous work [13], and authors claim that DG algorithm in [13] rely on an infinite horizon but this work consider a limited horizon. So the question related to the motivation is that why a limited horizon is better? As titled in [16], skills learning increases planning horizon, thus it seems that the horizon considered by a human might be variable and approaches to a very large number with time evolving.

2. To obtain the Eq. (1), despite the mentioned assumption, authors need to know the dynamics f . It seems that the dimension of the system is quite low from Eq. (10) and contents below it. However, it might be difficult to obtain an accurate f for much more complex systems, especially for the human-robot interaction cases. With an inaccurate dynamics f , the u_d could be misleading, and the linearized model will be inaccurate and less reliable. So, how about the robustness of the proposed algorithm for inaccurate dynamics? It is highly recommended to add experiments for robustness validation.

3. At line 92, should these Q matrices be at least semi-positive definite instead of being semi-positive definite?

4. Contents under Eq. (5): Is λ tuneable in [13]-[15]? Or Can the approaches presented in [13]-[15] be easily extended to a tuneable λ ? This relates to the fundamental novelty of this paper. Authors mentioned that including diverse behaviours in human partner is one of the key strengths of this work compared with previous works [13]-[15].

5. At Algorithm 1, line 2: Why $\tilde{\epsilon}(t)$ needs to be estimated? Above (9) authors mentioned it is measured from sensors. If it is indeed estimated, how do authors estimate it?

6. I find that there might exist a serious real-time issue for conducting the Algorithm 1. From line 3 to line 5, there seems exist two groups of iterations. One is to compute u regarding to the iterations (6), another is to compute θ with a local optimization algorithm. Authors need to clarify this point clearly and discuss the potential real-time issue.

7. The experimental validation lacks comparisons between the proposed approach and existing methods, such as [13]-[15].

8. How about the real-time issue in the experiments? Although authors give an explanation to reduce the influence of the bi-level optimization during line 422 to 457, more detailed settings and results are lacked in experiments.

Reviewer #3

(Remarks to the Author)

My comments have been well addressed.

Version 2:

Reviewer comments:

Reviewer #2

(Remarks to the Author)

Thanks for your reply. Although most of my concerns have been solved, there are still some major concerns remained.

For the reply of comment 1. The motivation for using finite horizon is still not strong enough by only introducing the results in [16]. Since this is the main strength of this paper compared with existing works. At least, comparisons between the approaches with finite and infinite horizons should be conducted. The reviewer has a great doubt about the disadvantages of infinite horizon.

For the reply of comment 2. To make the work self-contained, authors are encouraged to conduct experiments for robustness analyses by using biased model parameters.

For the reply of comment 4. Authors should reply to the comment 4 more straightly. The reviewer cares that whether the approach is just a simple extension of these previos works by allowing a tunable λ or originally designed a novel framework for considering the modulatable human efforts.

For the reply of comment 7. Comparisons are needed for better explanation. Authors said that previous methods cannot be used for the time-variant linear system and thus cannot be compared. However, for previous approaches they just used different modelling approaches and still could be used for the same experiemntal tasks conducted in this paper. The reviewer expect to see that the proposed approach can achieve obviously better performances comapred with existing approaches on the same human-robot collaboration tasks, especially on some points like using finite horizon and tunable human efforts.

Version 3:

Reviewer comments:

Reviewer #2

(Remarks to the Author)

I have no further comments now.

Answer to reviewers: Model predictive game control for personalized and targeted interactive assistance

We would like to thank the reviewers and editors for their constructive comments that contributed to greatly improve the manuscript. We provide below a point-to-point response to the editor’s and reviewer’s comments. Our answers appear in blue and the modifications in the revised manuscript are also highlighted in blue for the sake of clarity. We also provide a version of the manuscript without highlighted corrections. Line numbers provided in this answer are those in the files with corrections in blue.

1 Reviewer #2

This paper introduces a model-predictive game (MPG) controller for personalized physical human-robot interaction. A finite-horizon differential game framework uses receding-horizon Nash equilibrium to model human-robot co-adaptation, explicitly accounting for human limited planning horizon. Experimental validation with 30 participants demonstrates bidirectional adaptation. The robot identifies stable, individualized human control strategies, while humans adapt to the robot. Here are the comments on this paper.

Comment 1: Differential game theory is typically applied to non-cooperative adversarial scenarios. Please explain the motivation for utilizing this theoretical tool in controller design.

Differential game theory has indeed been applied to a large variety of adversarial scenarios. However, this same framework holds the potential to generate a continuum of dynamic interactions ranging from competition to integrated and user-specific cooperation as discussed in previous papers [1, 2]. While the term ‘non-cooperative’ may seem misleading at first, this framework readily allows to consider two agents collaborating and sharing control in a task. We added the following statement in the introduction to clarify this point (p.2, l.37):

*“[...] the concept paper [1] proposed modeling a contact robot and its human user as two agents with **their** own actuation, sensing and motion planning ability. ~~–and employing~~ **In this context**, differential games (DG) theory ~~to design their interaction control~~ **was suggested as a promising framework to flexibly adjust the interaction paradigm, with possibilities ranging from competition to cooperation (assistance) and design their interaction control accordingly.**”*

Comment 2: The paper merely mentions the limitations of traditional DG algorithms but provides no quantitative comparisons against mainstream human-robotic interaction control methodologies.

We acknowledge the reviewer’s comment. However, the main goals of this study were to (i) describe the MPG framework in details; (ii) test whether humans could adapt to a MPG-based controller, resulting in a consistent apparent Nash equilibrium; and (iii) assess whether humans could be guided towards other Nash equilibria by tuning the robot controller (as allowed by our λ meta-parameter adjusting the robot contributions in the task). Quantitative comparison of differential games-based control against some mainstream assistive controllers have already been addressed in previous works, e.g. comparison between linear quadratic control (LQ) and an infinite horizon DG [2] or comparison between model predictive control (MPC) and MPG [3]. A quantitative comparison with a variety of mainstream controllers is beyond the scope of this paper but the main theoretical differences are emphasized at the beginning of the introduction.

Comment 3: How does the combination of inverse DG, receding horizon, and λ -based behavior modulation distinguish MPG from prior work?

We thank the reviewer for this question which helps us to clarify the positioning of our work. The main differences between our method and previous ones rely in the fact that previous methods cannot verify the three properties ("a", "b" and "c") that we provide in the first paragraph of the introduction, whereas MPG can. We have added some clarifications in the Introduction regarding how MPG addresses "c" (p.2, l.47):

*“Here we develop a model predictive game (MPG) controller for a contact robot that continuously infers the human motor control on a finite horizon [3], i.e. with properties (a) and (b). To address (c), we incorporate a homotopy-based mechanism with an assistance meta-parameter that regulates the **robot’s internal objectives in terms of human effort minimization**. However, the co-adaptation between a human and a contact robot using a DG controller has not been thoroughly investigated before. **While previous studies successfully implemented DG-based assistance controllers inducing stable interaction between human and robot agent [2, 3], co-adaptation between the two agents has yet to be more thoroughly investigated.**”*

Comment 4: The methodology description focuses exclusively on optimization objectives yet omits detailed specifications of optimization constraints. Critical practical implementation considerations, such as bounded control inputs, remain unaddressed.

We agree that some details regarding the optimization constraints were overlooked. First, the boundaries used for the maximal/minimal cost weights (critical for robot stability), the maximum robot torque during online control (critical for safety), and the boundaries for the identification of the human cost parameters (important for the bi-level optimization) are now provided in the following statements in the Methods (p.16, l.416):

*“The robot cost was $\mathbf{Q}_r = \text{diag}(30, 0.1)$ for all experiments and all participants, **which was well below the stability boundaries evaluated in simulation as $\text{diag}(25500, 25)$. In addition, the applied robot torque, first computed through Eq. 7, was then saturated at 2 Nm for safety reasons and to comply with hardware constraints.**”*

Second, we acknowledge that the current method we use for the estimation of the human cost function is based on the assumption of non-bounded human control inputs. We agree with the reviewer that lifting this assumption could be more realistic and may impact the identified values of $\hat{\theta}$, in particular if boundary values are reached in the optimal solution. Even though we were not confronted to this scenario in our experiment (all variables remained below physiological bounds with our healthy participants), we now discuss this specific point in the Discussion, with the following statement and reference (p.14, l.363):

“Furthermore, the present methodology relies on the assumption that control inputs and state variables are unbounded. Additional work would be needed to relax this assumption by considering methods that allow control/state constraints in the direct DG solving and that possibly seek to identify both the human cost function and control boundaries, which could be important with pathological participants (e.g. [4]).”

Comment 5: EMG signals are recorded but minimally analyzed. How does EMG inform the human effort model or could refine the inverse DG optimization?

In the present paper, EMG data were only analyzed as a confirmation of the trends observed in the torques estimated using inverse dynamics. Importantly, Fig. 5F shows minimal activity of the wrist flexors and trends of extensors activity that are consistent with torque estimations. For instance, this demonstrates that their was very limited effort provided by the users under the form of co-contraction (which cannot be captured by inverse dynamics). To clarify this, we added the following statement in the Results (p.11, l.241):

“Importantly, beyond this reduction of extensors’ activity, we did not observe any change in co-contraction—that is a simultaneous activation of both flexors and extensors resulting in increased joint stiffness without generating movement. This confirms that the decreased net joint torque obtained through inverse dynamics did not mask an increased co-contraction effort.”

Furthermore, although we agree that EMG signals could theoretically be used as a component of the MPG control scheme, in particular to improve the human torque estimate via EMG-to-torque models, this was out of the scope of the present study and induces a number of complications in view of in-field applications. These complications were made clear in the Discussion by extending previous statements (p.12, l.289):

“However, transmission latencies and filtering issues inherent to EMG processing drastically limit the usable time horizon [5], rendering such methods impractical for providing a truly proactive assistance. Furthermore, the in-field implementation of EMG-based assistive controllers is also hampered by practical limitations, such as the complex and time consuming process of installing the sensors and calibrating the EMG-to-torque model [6]. In addition, EMG signals are noisy and require prior filtering, which is time consuming in real time control scenarios. Finally, the quality of EMG signals tends to decrease through time due to environmental conditions, such as sweat between the skin and the electrodes. Yet, EMG could be useful to estimate physiological parameters such as fatigue, which could in turn provide relevant information to modulate the λ parameter in MPG.”

Comment 6: Does the discussion of model parameters sufficiently characterize complex human behaviors? Is it necessary to incorporate additional physiological indicators?

Analyzing human behavior often requires several metrics. Here we have focused on human cost parameters because of the assessment of the MPG method. However, beyond the model parameters, we have analyzed (i) the task-errors for different conditions; (ii) the net human joint torque estimated from inverse dynamics; (iii) EMG activities of the muscles involved in the task; and (iv) the synchronization of the human with the robot through time (correlation analyzes of human and robot torques). Therefore, we have provided a relatively complete description of the effects of MPG on human motor behavior for our task. We agree with the reviewer that for more complex tasks, in particular involving multiple joints, it would be possible to use more advanced indicators such as human inter-joints coordination or muscle synergies.

Comment 7: The experiment participants consist exclusively of "young, right-handed adults," without validation across diverse populations—such as individuals of varying ages or pathological conditions like motor impairments.

We thank the reviewers for this comment. It should be noted that despite the relative homogeneity of our tested population at first sight, their identified behaviors are clearly different, for instance going from 40 to 100 in the cost associated with tracking the trajectory. It shows that healthy people can spontaneously adapt to various Nash equilibria. Furthermore, the preliminary analysis of the method outcomes with passive and active participants show that MPG can reliably assist users that are not contributing to the task, as could be the case in presence of muscle weakness. This passive condition could be seen as simulating an individual with severe muscle weakness. These initial analyses were necessary to show the robustness of our method in assisting a variety of individuals. However, we agree with the reviewer that it would be interesting to test our method would with impaired individuals, such as post-stroke patients where spasticity could cause some issues. We hypothesize that our MPG controller would identify any spasticity-induced resistance as a null cost function parameters and will thus increase assistance. Whether this can generate oscillations or prevent the robot from identifying a stable human motor strategy remains to be investigated in future works. This limitation is now made clear in the Discussion (p.13, l.319):

“Importantly, we have shown that this biomarker can be identified for behaviors ranging from passivity to full engagement in the task performance. However, the present study does not address what would happen in impaired individuals, for instance in post-stroke patients with spasticity that could result in more complex interactions. This important limit could be addressed in simulation studies, e.g. showing that increased motor noise or viscoelasticity does not prevent the method from identifying the human costs [7], and in clinical studies with impaired patients.”

Comment 8: The MPG approach relies on the online solving of differential games, yet the paper fails to report computation time metrics or validate the real-time feasibility of the control algorithm.

It is indeed important to give time metrics to estimate the real-time feasibility of the method. We did mention that we limited the number of iterations of the bi-level optimization method so as to allow real-time control of the robot (in that case at 100 Hz or 10 ms per iteration, see Methods: p.17, l.456) without hampering the convergence of the identification method and the effective sharing of task efforts. Importantly, it should be noted that those choices were constrained by (i) the control rate of the robot, 100 Hz as mentioned; (ii) the programming language of the robot (Matlab), whereas other languages such as C++ are a lot faster (typically <1 ms per iteration for the same operations); and (iii) the USB connection of the device to the computer, that is much slower than Ethernet connections broadly used in robotics. Therefore, our experiments demonstrate the real-time applicability of the method in our settings. Note that

time metrics are dependent on the used computer, so providing raw iteration times would not be very informative in the present study as they would not be compared to other controllers or identification methods implemented on the same platform. For more complex robots with more degrees of freedom, real-time feasibility could be more challenging, but solving the differential equations can be very fast and it may not be necessary to update the controller via bi-level optimization at 1kHz since we predict the human torque on a receding horizon.

Comment 9: Validation is limited to 1-DOF wrist control. How does the proposed method address scalability to multidimensional systems?

This is indeed an important question. Importantly, although our experiments focused on a 1-DoF task to investigate human adaptations to the controller, the MPG framework introduced in the present study is formulated in the general case, so that all the equations can be directly applied to a n -joints system with $2n$ state dimension. We have revised the Discussion to include the following reference showing the scalability of our method through 2-DoF simulations with noises and errors (p.14, l.369). Furthermore, in its current form, the method requires a reference trajectory-control pair to be given to perform the linearization.

“Finally, while we have shown in a recent simulation study that the present method is scalable to multi-joint cases in principle [7], future works will need to investigate whether humans can adapt to MPG-controlled robots in multi-joint tasks, where human-robot coordination may be more difficult to learn.”

Comment 10: The recency of the references is insufficient. Please incorporate citations of state-of-the-art advances from the last three years.

We thank the reviewer for this suggestion and have added state of the art references to improve our Introduction (p.2, l.33) and Discussion (p.12, l.282). We now cite Xiong et al. on multifunctional elbow exoskeleton control [8], Zhao et al. on adaptive control strategies for wrist exoskeletons [9], Othman and Yang on human-robot collaboration for manufacturing [10] and Mehrholz et al. for a recent meta analysis on the outcomes of robot-assisted rehabilitation [11].

2 Reviewer #3

The major claims of the paper is a model predictive game (MPG) control for physical human-robot interaction (pHRI) control. The current version is suggested as major revision. Please check following comments

Comment 1: The MPG is novel and will be of interest to others in the community and the wider field. However, its novelties has not yet been highlighted by comparing state-of-the-art(SOTA) such as adaptive impedance control [1, 2] in exoskeleton experiments.

[1] Zhao, Yihui, et al. "Adaptive cooperative control strategy for a wrist exoskeleton using model-based joint impedance estimation." IEEE/ASME Transactions on Mechatronics 28.2 (2022): 748-757.

[2] X. Xiong et al. "Learning-based multifunctional elbow exoskeleton control." IEEE Transactions on Industrial Electronics 69.9 (2021): 9216-9224.

We thank the reviewer for these suggestions. We have added the suggested references in the Discussion, alongside other references on adaptive impedance control already mentioned (line 282). Importantly, as we mention in the Discussion, the critical difference between such methods and the proposed MPG is that they cannot plan the assistance, limiting them to reactive assistive control, whereas MPG is a proactive assistive method which decides its current action based on a prediction of the upcoming human behavior. This critical distinction relates to the fact that impedance control does not consider the human as an independent intelligent agent, with their own planning capacities, but only provides an assistance scaling with actual or past performance.

Comment 2: The conclusions and discussion are incomplete, it would be helpful if you could add references and contents on i) the advantages and extensions of the proposed MPG over neural network based control and adaptive impedance control in multi-joints [3, 4].

[3] Li, Sitan, and Chien Chern Cheah. "Deep Neural Network-Based Jacobian Control of Robot Manipulators: Offline Regression and Online Adaptation." IEEE/ASME Transactions on Mechatronics (2025).

[4] "Online sensorimotor learning and adaptation for inverse dynamics control." *Neural Networks* 143 (2021): 525-536.

We agree that we did not discuss neural-network-based alternatives properly. We have now added a statement and a reference (we chose a reference closer to our targeted application to exoskeletons) in the Discussion to address this point (p.14, l.353):

"[...] This large panel of possible interaction scenarios further allows the MPG framework to be more flexible in setting the robot's assistive contribution than recent alternatives, for instance based on reinforcement learning [12] or on adaptive impedance control [8]. Interestingly, future works could focus on combining learning-based approaches and MPG, in particular by using reinforcement learning for the estimation of the human cost function. Such approaches could be very powerful to mitigate the curse of dimensionality when solving the inverse problem in a multi-joints setting."

Comment 3: the contribution of the proposed should be clearly stated by comparing SOTA [1,2,4]. Adding a comparison table should be helpful.

This is related to the previous reviewer's comments but adding a comparison table is not straightforward because there is no direct metric that we could use to compare our method to adaptive impedance control or neural-network-based approaches. Importantly, we already emphasized the fundamental differences between the principle of adaptive impedance control and MPG in the Introduction and the Discussion (as mentioned in answer to the reviewer's comment 1). Regarding the neural-network based control, we have now included a statement (see answer to the reviewer's comment 2) and references to discuss (i) why such controllers (e.g. based on reinforcement learning [12]) are currently less flexible than our approach; and (ii) how such learning methods could be useful to improve the real-time identification of the human cost function, in particular to mitigate the curse of dimensionality.

Comment 4: A multi-jointed wearable robot was included in Fig. 1, but the validation is based on a one-DOF exoskeleton in Fig.2. It is misleading.

Figure 1 aims at describing the concept of the MPG framework as stated in the caption, and is therefore consistent with the general formulation of the equations for a n -DoF system as they are provided throughout the paper. On the other hand, Figure 2 clearly shows the 1-DoF experimental setup before presenting any experimental result. Therefore, we believe that the difference between (i) the general n -DoF theoretical part (illustrated in Fig. 1) and (ii) the 1-DoF experimental part to study human adaptations to MPG (illustrated in Fig. 2) is relevant for this paper. However, we have included and additional clarification in the caption of Figure 2 "*Control scheme and experimental task **with 1-DOF exoskeleton.***"

3 Reviewer #4

3.1 Introduction

Comment 1: The abstract and introduction mention "assistance meta-parameter," but the mechanism and implications are not clearly intuitive until much later (or just mentioned a little in Figure 1). Consider previewing how this parameter works earlier for better reader orientation.

We thank the reviewer for pointing out that the mechanism associated with this meta-parameter was not clear enough early in the paper. We added the following details (in bold):

- **Abstract** (p.1, l.17): "*[...]their behavior can be modulated through an assistance meta-parameter **adjusting the robot's propensity to minimize human effort.***"
- **Introduction** (p.2, l.48): "*To address (c), we incorporate a homotopy-based mechanism with an assistance meta-parameter that regulates the robot's internal objectives **in terms of human effort minimization.***"

Comment 2: The authors state that "co-adaptation between a human and a contact robot using a DG controller has not been thoroughly investigated before." This is a strong claim. If possible, temper it by acknowledging adjacent studies or explaining what makes their investigation distinct.

We agree with this comment and have rephrased the sentence in the Introduction (p.2, l.48):

“ [...] an assistance meta-parameter that regulates the human’s and robot’s involvement **by adjusting a compromise between the two agents’ effort costs**. However, ~~the co-adaptation between a human and a contact robot using a DG controller has not been thoroughly investigated before.~~ **While previous studies successfully implemented DG-based assistance controllers inducing stable interaction between human and robot agent [2,3], co-adaptation between the two agents has yet to be more thoroughly investigated.** Would users actively contribute to the movement even when the robot could [...]”

3.2 Results

3.2.1 Interaction control as differential game on a receding finite horizon

Comment 3: Consider adding brief intuitive descriptions alongside the equations to improve accessibility for the readers unfamiliar with control theory. (For example, ξ represents deviation from the planned trajectory, which the robot attempts to minimize...)

We thank the reviewer for this comment that helped us make the paper more accessible. We have added the following statements in the Results (in bold):

- (p.3, l.81): “[...]This affine system of equations describes how the human and robot inputs affect ξ , **the deviations from the planned trajectory that the agents will attempt to minimize.**”
- (p.4, l.95): “In above equations, $\mathbf{Q}_r, \mathbf{Q}_r^f, \mathbf{Q}_h, \mathbf{Q}_h^f \in \mathbb{R}^{n \times n}$ are positive semi-definite matrices weighting task error minimization, **which increase the amount of effort that agents can expand to reduce tracking errors**, while $\mathbf{R}_r \in \mathbb{R}^{m \times m}$ and $\mathbf{R}_h \in \mathbb{R}^{m \times m}$ are positive definite matrices weighting effort minimization, **resulting in a compromise between tracking accuracy and effort expenditure.**”
- (p.4, l.98): “The positive semi-definite matrix $\mathbf{R}_{rh} \in \mathbb{R}^{m \times m}$ reflects the extent to which the robot aims to assist the human partner. **Note that the human cost function does not include a $\mathbf{R}_{hr} \in \mathbb{R}^{m \times m}$ term as there is no reason to assume that humans would attempt to minimize the partner’s effort in our robotic assistance setup. However, such a term could appear during interactions with a social component, such as between humans or with social robots.**”

Comment 4: It would be helpful to explain why the linearization around a planned trajectory (as in Eq. 1) is valid in practical scenarios.

We agree with the reviewer that some information was missing on this critical point. We added the following statement in the Results (p.3, l.81):

“In practice, this linearization is valid when deviations from the desired trajectory remain small. This assumption is plausible in a variety of human-robot interaction scenarios, either because (i) both agents are provided with the desired trajectory and instructed to track it, e.g. during a rehabilitation protocol; or (ii) because the robot is able to extract an accurate estimation of the desired trajectory of the human, e.g. during an industrial task. In general, the method requires the knowledge of both a reference state and control trajectory, a process which may require leveraging other methods (e.g. [13]), but this is out of the scope of the present paper and left for future work.”

Comment 5: It would help to explicitly state or remind the reader what the intuition is behind the backward-in-time integration (Eq. 6).

The backward-in-time integration is classical to solving LQR optimal control problems via Riccati equations. To give some insights, we rephrased the sentence mentioning it as follows (p.4, l.112):

“The optimal control for each agent can be obtained by integrating a set of ordinary differential equations with unknown matrices and vectors $\mathbf{P}_h(t), \alpha_h(t), \mathbf{P}_r(t), \alpha_r(t)$ backward in time. First, we initialize the value of these matrices at the end of the planning horizon $\mathbf{P}_i(\tau_f) = \mathbf{Q}_i^f, \alpha_i(\tau_f) = \mathbf{0}, i \in \{r, h\}$. Then the following equations are iteratively integrated backwards in time, which is by starting from $t = \tau_f$ and finishing with $t = \tau$ the current time, to obtain the time evolution of the unknown matrices over the planning horizon Δ_p .”

3.2.2 Inverse DG for human cost estimation

Comment 6: A brief justification of why diagonal matrices are assumed for Q_h, Q_h^f, R_h would be useful.

To address this comment, we have rephrased the sentence introducing the cost matrices as follows (p.5, l.128):

“For this purpose, we use diagonal matrices Q_h, Q_h^f, R_h , which correspond to the assumptions that humans (i) correct errors in position and velocity independently along each task dimension, and (ii) minimize control costs independently for each task dimension. Both these assumptions are plausible in humans and have allowed seamless human-robot interactions using infinite horizon DG control [2].”

Comment 7: For the cost function in Eq. (9) it may be useful to discuss whether the problem is convex or how optimization is handled numerically, especially since θ appears in a nonlinear fashion through the solutions of Eq. 6 and 8.

We agree with the reviewer that details were missing regarding the discussion of the problem’s convexity and the numerical optimization. Therefore, we added the following details:

- **Results** (p.6, l.145): *“[...] semi-definite matrices weighting the control and state errors. Note that computing $\delta u_{h,\theta}$ requires solving coupled ordinary differential equations (see eq. (6)) making convexity guarantee of the problem difficult. For practical efficiency, we employ a local optimization algorithm presented in the Methods section.”*
- **Methods** (p.16, l.442): *“We used Powell’s Bound Optimization BY Quadratic Approximation method (BOBYQA), that employs an iterative quadratic approximation of the objective function in Eq. (9), then minimizes it using the conjugate gradient procedure [14]. It is a local, derivative-free optimization method that proved to be suited and robust for our purpose in preliminary simulation studies.”*

Comment 8: Can you discuss a little bit about the robustness or sensitivity for the inverse DG and bi-level formulation?

We thank the reviewer for this suggestion. We have added the following statement mentioning previous robustness studies of our bi-level approach in the Methods (p.16, l.445):

“In simulation works, we have shown that our method can robustly identify correct human costs during 1-DoF movements with respect to different initial guesses [3] and with respect to noise/modeling errors in 2-DoF movements [7].”

3.2.3 Experiments

Comment 9: All three experiments rely heavily on the interpretation of θ_1 as a proxy for human intent or engagement. It would strengthen the narrative to briefly justify this assumption (e.g., why θ_1 is a valid or sufficient proxy) early in the section or as a transition to Experiment 1.

We agree with the reviewer’s comment. Indeed, θ_1 is a good proxy to quantify the human involvement in the task and predict their upcoming behavior: a passive participant or participant who does not execute the task should lead to a small θ_1 value, where as an active participant who attempts to track the desired trajectory should lead to a larger value of θ_1 . Therefore, we added the following statement to clarify this aspect in the “Experiments” section (p.6, l.160):

“Furthermore, whether it be for the validation of the method or to study human involvement and adaptation, we will use the identified human position error weight $\hat{\theta}_1$. First, this parameter can be used to verify the validity of the method by measuring human effort and performance and analyzing whether they are consistent with the identified value. For instance, if \tilde{u}_h is null, i.e. the human is passive, then the identified $\hat{\theta}_1$ should be close to zero, which we test in Experiment 1. Second, once the consistency of $\hat{\theta}_1$ with the human behavior is established, it can be used as a proxy for human involvement and performance in the task as in our Experiments 2 and 3, a high $\hat{\theta}_1$ corresponding to a user accurately tracking the desired trajectory.”

Comment 10: It would be helpful to include whether assumptions of normality were tested, especially for ANOVAs or Friedman tests.

We tested for normality, which assumption was not verified in our dataset. That is why we used Friedman tests for main effects and Wilcoxon signed-rank tests for post-hoc comparisons. Indeed, these tests are non-parametric, which means that they do not assume normality. To make this clear in the paper, we have modified the description of our statistical processing as follows (p.18, l.487):

“We first conducted a Shapiro-Wilk test to verify whether our data was normally distributed. This test showed that our data violated the assumption of normality. Therefore, to examine the effects of experimental conditions on interaction metrics, a non-parametric Friedman test was performed with effect sizes reported as Kendall’s W for significant outcomes. For pairwise comparisons of conditions, non-parametric Wilcoxon signed-rank tests were conducted, and the effect sizes were reported as the rank-biserial correlation r . Statistical significance was set at $p < 0.05$, and adjustments for multiple comparisons were made using the Bonferroni method.”

Comment 11: For Experiment 1, you might consider including effect size (r) for the non-significant results on θ_2 to contextualize their potential practical (vs. statistical) insignificance.

We thank the reviewer for this suggestion. We added detail about the effect size observed in the non significant results (p.7, l.187):

“[...] between the active and passive conditions with medium effect size ($p > 0.05$, $r=0.267$).”

Comment 12: For Experiment 2, the decomposition method for variance is not described. A brief mention or reference in the Methods section would help readers replicate or verify this analysis.

We thank the reviewer for pointing out this oversight. We added the following statement to clarify this point in the “Data Analysis” section (p.18, l.494):

“The decomposition of variance used to analyze the consistency of the apparent Nash equilibria was based on the sum-square (SS) output of the Friedman test:

$$V_{inter} = \frac{SS_{inter}}{SS_{inter} + SS_{intra}} \times 100, \text{ and } V_{intra} = \frac{SS_{intra}}{SS_{inter} + SS_{intra}} \times 100 \quad (1)$$

”

Comment 13: The fact that CI increases while its variability decreases is a nice indicator of co-adaptation. Consider highlighting whether this improvement is more strongly driven by changes in the human or robot strategy.

We thank the reviewer for this interesting comment. During each trial, the inverse differential game method takes a few seconds to converge to a plateau as illustrated by the steep curve in the identified $\hat{\theta}_1$ at the beginning. This part of the trial mainly reflects the robot’s identification of, and adaptation to, human behavior, while possibly also including some human adaptation. Then, the changes in the reached plateau between trials, that correlate with changes in CI, can mainly be attributed to the human adapting their strategy to the assistance of the robot. Specifically, as the robot largely provides the effort necessary to complete the task, users can focus on correcting tracking errors, triggering an increase in $\hat{\theta}_1$ (see Fig. 4A). Furthermore, it should be noted that the robot strategy (i.e. its cost function) in itself is fixed for Experiment 2. This means that changes observed beyond the human identification phase are most likely due to the human, as mentioned above.

3.3 Discussion

Comment 14: The discussion of EMG-based model predictive control ([29], [30]) could be improved by clarifying what specifically limits its practical use.

We agree with the reviewer that, although we already discuss the real-time computation problem associated with these methods, other practical limits can be discussed. Therefore, we added the following sentence in the “Discussion” (p.12, l.289):

“Furthermore, the in-field implementation of EMG-based assistive controllers is also hampered by practical limitations such as the time consuming process of installing the sensors and calibrating the EMG-to-torque model [6]. Finally, the quality of EMG signals tends to decrease through time due to environmental conditions, such as sweat between the skin and the electrodes.”

Comment 15: When discussing user engagement despite no explicit instruction (e.g., in Experiment 3), consider noting any observed variation between participants.

While we have presented the participants inter and intra-variability analysis in experiment 2, showing that the MPG controller allows estimating a personalized cost function parameters, we agree with the reviewer on the necessity of presenting similar variability analysis between participants in the third experiment. We have added the following analyses in the Results and Methods sections:

- **Results** (p.11, l.246): “We then analyzed whether the apparent Nash equilibrium exhibited by individuals was robust across different values of λ , which would show a consistency of the user-specific assistance provided by MPG, even for different targeted human and robot contributions. To test this premise, we computed the average $\hat{\theta}_1$ across the population for each λ , and the distance to this population average for each participant and each λ . Then, we conducted Pearson correlation tests to analyze whether an individual’s distance to the population average was robust across conditions. These tests showed that the identified individual strategies were consistently ranked across participants and conditions for $\lambda \neq 1$ ($\lambda = 0$ vs 0.5 and 0.7: $r > 0.5$, $p < 0.015$). However, the participants’ behavior with $\lambda = 1$ were not correlated to their behavior in the other conditions (all comparisons: $r < 0.11$, $p > 0.33$). With the same analysis strategy, a similar pattern was observed in participants’ effort contributions (i.e. average torque): conditions with $\lambda < 1$ were correlated (all comparisons: $r > 0.5$, $p < 0.015$), while the condition $\lambda = 1$ was not correlated to other conditions (all comparisons: $r < 0$, $p > 0.08$). These results suggest that $\lambda < 1$ gives more flexibility to participants, allowing them to express user-specific differences. Conversely, when $\lambda = 1$, the robot’s contribution to the task is largely aimed at reducing human inputs, resulting in an homogenization of behaviors as the robot takes the lead while the human becomes more passive.”
- **Methods** (p.18, l.497): “To test the robustness of the individual apparent Nash equilibrium, we conducted pairwise Pearson rank correlation tests on the difference between population average behaviors and individual behaviors between different conditions. We report the Pearson correlation coefficient r and the corresponding p -values.”

Comment 16: The paragraph could be improved by explaining how close human behavior actually comes to MPG predictions.

We thank the reviewer for the suggestion. We have improved the paragraph by adding the following considerations (in bold) in the Discussion (p.13, l.329):

“[...] Do humans interacting with a robot behave as predicted by differential game theory? Although [15, 16] provided some evidence that human motion aligns with a Nash equilibrium after learning, the online control strategy employed by humans may significantly differ from the MPG algorithm which is used as a tool to control the robot. **In fact, the average RMSE between the measured and predicted participants’ torque was ≈ 0.3 Nm, which was mostly due to an under-estimation of the human control inputs. This under-estimation likely results from the under-estimation of errors induced by the human as a consequence of sensorimotor noises, as they are not accounted for by the human model. To overcome this limit, the human model may need to be formulated as a stochastic optimal control problem so as to account for this noise [17] and external uncertainty [18], e.g. related to the robot. Nevertheless, the fact that the participants skillfully adapted their behavior to the robot’s commands—whether it be through learning in the second experiment or through different robot contributions in the third experiment—indicates that they understood the robot strategy and considered it to control their movement. In this sense, our results offer compelling evidence that the human interactive behavior is controlled in game theory like manner.**”

Further results of the comparison between the predictive torques and the measured torques were added in the supplementary file:

“Here, we present supplementary results that show how close is MPG prediction to the actual human behavior. Additional analysis were conducted on data from experiment 2, where we calculated the root mean square error (RMSE) between the

measured human torque \tilde{u}_h and the MPG predicted torque \hat{u}_h noted \tilde{u}_{hRMSE} ".

Figure 1: Measured participants torque vs MPG estimated torque. Results are displayed over consecutive MPG trials of experiment 2. **A.** Root mean square error of the MPG estimated participants applied torques u_{hRMSE} during the first quarter of the trials. **B.** Root mean square error of the MPG estimated participants applied torques u_{hRMSE} during the last quarter of the trials. **C.** Coordination index between the participants measured torques and the MPG torques during the last quarter of the trials.

“Fig. 1A shows the error during the first 30 second of the trials where the average error between all trials was $\bar{u}_{hRMSE} = 0.43$ Nm. No significant difference was observed between the trials ($p = 0.68$). Fig. 1B shows the error during the last 30 second of the trials where the average error was $\bar{u}_{hRMSE} = 0.31$ Nm. No significant difference was observed between the trials ($p = 0.22$). However, the decrease of the error between the first and the last 30 seconds of the trials was statistically significant ($p < 0.001$, $r > 0.9$). This results suggest that the prediction improves as the estimation of the participants cost parameter reaches the Nash equilibrium plateau independently of the human adaptation across trials. Fig. 1C shows the evolution of the coordination index during the last 30 seconds between the measured participants toques and the MPG predicted torques across the trials. Where the coordination index significantly increased across trials ($p < 0.05$ Kendall’ $W = 0.02$, $\chi_F^2 = 17.08$, post-hoc test showed significance for trial 1 vs trial 3 and trial 5, with $p < 0.05$).”

3.4 Methods

Comment 17: Consider adding a brief overview paragraph before the detailed breakdown to help the reader mentally organize the coming subsections.

We thank the reviewer for this comment that helped us improve the readability of our manuscript. We added the following paragraph at the beginning of the method section (p.14, l.376):

“In this section, we first provide details regarding the experimental setup and task used for the three experiments, followed by the resulting human-robot interaction dynamics. Then, we detail the implementation of the MPG controller and of the bi-level identification for our specific interaction dynamics. Finally, we provide the anthropometric data of the tested participants and details of the protocol for each of the three experiment, followed by the employed data processing techniques and statistical tests.”

Comment 18: Please clarify whether the sample size for each experiment was determined based on a power analysis.

The sample size was not determined through a power analysis. We chose the number of participants based on usual numbers in the literature and on the number of conditions investigated in each experiment. For instance, since only two conditions were treated in Experiment 1 for the validation of the method, fewer participants were needed. However, we have now added a statistical power analysis *a posteriori* using G*power. In all tests but one, the power level was > 0.80 and this is now clarified in the Results (p.11, l.264):

“[...] and slightly decreased the velocity error ($p < 0.001$, $W = 0.17$, $\chi_3^2 = 20.46$, with pairwise significant difference only between $\lambda = 0$ and $\lambda = \{0.7, 1\}$, $p < 0.05$). **Note that in this test the statistical power was 0.60. For all the other reported tests of the present paper, the statistical power was > 0.8 .**”

We also added the following details in the Methods (p.18, l.493):

“Finally, we performed a posteriori power analyses for all the reported statistical tests using G-power, which showed that all but one tests had a high power > 0.8 .”

Comment 19: Consider adding a sentence justifying why gradient-free methods are preferred over gradient-based ones in the BOBYQA setup.

We agree with the reviewer that this point was not clearly justified. Therefore, we added the following statement in the “Real-time bi-level optimization” section (p.16, l.449):

“In our context, gradient-based algorithms led to less stable convergence of the human costs estimation, which is why gradient-free ones were privileged.”

References

- [1] N. Jarrassé, T. Charalambous, and E. Burdet, “A framework to describe, analyze and generate interactive motor behaviors,” PLOS ONE, vol. 7, no. 11, p. e49945, Nov. 2012.
- [2] Y. Li, G. Carboni, F. Gonzalez, D. Campolo, and E. Burdet, “Differential game theory for versatile physical human–robot interaction,” Nature Machine Intelligence, vol. 1, no. 11, p. 36–43, Jan. 2019.
- [3] A. Hafs, D. Verdel, E. Burdet, O. Bruneau, and B. Berret, “A finite-horizon inverse differential game approach for optimal trajectory-tracking assistance with a wrist exoskeleton,” in 2024 10th IEEE RAS/EMBS International Conference for Biomedical Robotics and Biomechanics (BioRob), 2024, pp. 450–456.
- [4] Z. Chen, T. Bačėk, D. Oetomo, Y. Tan, and D. Kulić, “Inverse optimal control for dynamic systems with inequality constraints,” IFAC-PapersOnLine, vol. 56, no. 2, pp. 10 601–10 607, 2023.
- [5] L. Quesada, D. Verdel, O. Bruneau, B. Berret, M.-A. Amorim, and N. Vignais, “EMG feature extraction and muscle selection for continuous upper limb movement regression,” Biomedical Signal Processing and Control, vol. 103, p. 107323, May 2025.
- [6] —, “EMG-to-torque models for exoskeleton assistance: a framework for the evaluation of in situ calibration,” bioRxiv, Jan. 2024.
- [7] A. Hafs, D. Verdel, O. Bruneau, and B. Berret, “Game-Theoretic Interaction Control for Assistive Exoskeletons: a 2-DOF Simulation Study,” in IFAC 2025 - Joint 10th IFAC Symposium on Mechatronic Systems and 14th Symposium on Robotics, Paris, France, Jul. 2025. [Online]. Available: <https://hal.science/hal-05063077>
- [8] X. Xiong, C. D. Do, and P. Manoonpong, “Learning-based multifunctional elbow exoskeleton control,” IEEE Transactions on Industrial Electronics, vol. 69, no. 9, p. 9216–9224, Sep. 2022.
- [9] Y. Zhao, K. Qian, S. Bo, Z. Zhang, Z. Li, G.-Q. Li, A. A. Dehghani-Sani, and S. Q. Xie, “Adaptive cooperative control strategy for a wrist exoskeleton using model-based joint impedance estimation,” IEEE/ASME Transactions on Mechatronics, vol. 28, no. 2, p. 748–757, Apr. 2023.
- [10] U. Othman and E. Yang, “Human–robot collaborations in smart manufacturing environments: Review and outlook,” Sensors, vol. 23, no. 1212, p. 5663, Jan. 2023.
- [11] J. Mehrholz, A. Pollock, M. Pohl, J. Kugler, and B. Elsner, “Systematic review with network meta-analysis of randomized controlled trials of robotic-assisted arm training for improving activities of daily living and upper limb function after stroke,” Journal of neuroengineering and rehabilitation, vol. 17, pp. 1–14, 2020.
- [12] S. Luo, M. Jiang, S. Zhang, J. Zhu, S. Yu, I. Dominguez Silva, T. Wang, E. Rouse, B. Zhou, H. Yuk, X. Zhou, and H. Su, “Experiment-free exoskeleton assistance via learning in simulation,” Nature, vol. 630, no. 8016, pp. 353–359, 2024.

- [13] A. Orhan, D. Verdel, O. Bruneau, F. Geffard, and B. Berret, “Combining Model-based and Data-based approaches for online predictions of human trajectories,” in IEEE RAS EMBS 10th International Conference on Biomedical Robotics and Biomechatronics (BioRob 2024), Heidelberg (Germany), France, Sep. 2024. [Online]. Available: <https://hal.science/hal-04679979>
- [14] M. J. D. Powell, “The BOBYQA algorithm for bound constrained optimization without derivatives,” Department of Applied Mathematics and Theoretical Physics, Cambridge University, Cambridge, UK, Tech. Rep. NA2009/06, 2009.
- [15] D. A. Braun, P. A. Ortega, and D. M. Wolpert, “Nash equilibria in multi-agent motor interactions,” PLOS Computational Biology, vol. 5, no. 8, p. e1000468, Aug. 2009.
- [16] V. T. Chackochan and V. Sanguinetti, “Incomplete information about the partner affects the development of collaborative strategies in joint action,” PLOS Computational Biology, vol. 15, no. 12, p. e1006385, Dec. 2019. [Online]. Available: <http://dx.doi.org/10.1371/journal.pcbi.1006385>
- [17] B. Berret and F. Jean, “Stochastic optimal open-loop control as a theory of force and impedance planning via muscle co-contraction,” PLOS computational biology, vol. 16, no. 2, p. e1007414, 2020.
- [18] B. Berret, D. Verdel, E. Burdet, and F. Jean, “Co-contraction embodies uncertainty: An optimal feedforward strategy for robust motor control,” PLOS Computational Biology, vol. 20, no. 11, p. e1012598, 2024.

Answer to reviewers: Model predictive game control for personalized and targeted interactive assistance

We thank the reviewers and editors for their constructive comments, which we address point-to-point below. Our answers are provided in blue, and the corresponding modifications in the manuscript are also highlighted in blue. For convenience, we also provide a clean version of the manuscript without highlights. All line numbers referenced in this response refer to the version with corrections marked in blue.

1 Reviewer #2

Comment 1: It seems that the work is quite similar with the previous work [13], and authors claim that DG algorithm in [13] rely on an infinite horizon but this work consider a limited horizon. So the question related to the motivation is that why a limited horizon is better? As titled in [16], skills learning increases planning horizon, thus it seems that the horizon considered by a human might be variable and approaches to a very large number with time evolving.

In [13], the authors formulated the differential game problem in the infinite horizon setting (see their Eq. 4). In [14], the problem was initially presented in finite horizon (see their Eq. 3), but the authors ultimately let the horizon tend to infinity (remark 4), which was essential to derive their update rule (Eq. 15). Thus, the algorithm of [14] is effectively applied in infinite horizon as well. However, Bashford et al. [16] reported that in their experiment subjects adapt they planning horizon with practice, highlighting the importance of considering a finite horizon. The novelty of our algorithm lies in extending the method to the finite horizon case. Our motivation for this extension is twofold: (1) such methods require linearisation around the human motion plan, which must realistically be predicted on a relatively short time horizon; (2) linearisation of nonlinear systems introduces time-varying matrices in the affine approximation, making infinite horizon formulations unsuitable for solving LQ differential games.

Comment 2: To obtain the Eq. (1), despite the mentioned assumption, authors need to know the dynamics f . It seems that the dimension of the system is quite low from Eq. (10) and contents below it. However, it might be difficult to obtain an accurate f for much more complex systems, especially for the human-robot interaction cases. With an inaccurate dynamics f , the u_d could be misleading, and the linearized model will be inaccurate and less reliable. So, how about the robustness of the proposed algorithm for inaccurate dynamics? It is highly recommended to add experiments for robustness validation.

Thank you for this important question. Our algorithm in fact assumes that the dynamics f are known, which makes deriving Eq.1 essential. An example of this derivation is provided in Eq. 10. In ref. [37], we performed simulations to study the impact of inaccurate dynamics on a 2-DOF arm. Specifically, we tested uncertainties in inertia, damping and gravity. The results showed that the method is most sensitive to errors in the gravity model. We also evaluated robustness with respect to motor noise in the simulated human inputs. Furthermore, our experimental results confirmed that, despite uncertainties in human anthropometric parameters, the controller still provides effective assistance.

Comment 3: At line 92, should these Q matrices be at least semi-positive definite instead of being semi-positive definite?

We interpret the reviewer's question as whether the Q matrices should be positive definite rather than just semi-positive definite. According to Corollary 6.5 in [18] (p. 323), the Q matrices can be semi-positive definite. The matrix R , however, must be positive definite, as noted at line 94. Note that R_{rh} can be semi-positive definite.

Comment 4: Contents under Eq. (5): Is lambda tuneable in [13]-[15]? Or Can the approaches presented in [13]-[15] be easily extended to a tuneable lambda? This relates to the fundamental novelty of this paper. Authors mentioned that including diverse behaviours in human partner is one of the key strengths of this work compared with previous works [13]-[15].

References [13] and [15] did not include human effort in the robot objective, which is necessary to define a lambda based tuning of the relative effort contributions of human and robot. Music et al. [14] did include human effort in the cost function and could have implemented such lambda-based tuning, but just used a fixed lambda equivalent to 0.5 in our paper. We consider major contributions of our work (beyond the algorithm with finite horizon), to (i) recognising that this simple tuning mechanism provides a principled framework to regulate the human-robot relationship, and (ii) verify it in experiments with human participants that the robot can thereby impose a spectrum of relationships from master-slave to competition and independent co-activity.

Comment 5: At Algorithm 1, line 2: Why $\tilde{\xi}(t)$ needs to be estimated? Above (9) authors mentioned it is measured from sensors. If it is indeed estimated, how do authors estimate it?

In our setup, position was obtained from encoders and velocity via numerical differentiation, providing a basic estimate of $\tilde{\xi}$. More generally, any suitable estimator can be used under partial observation or for greater accuracy. Accordingly, we replace “measured from sensors” with “estimated from sensors measurements” (p.6, l.141).

Comment 6: I find that there might exist a serious real-time issue for conducting the Algorithm 1. From line 3 to line 5, there seems exist two groups of iterations. One is to compute u regarding to the iterations (6), another is to compute theta with a local optimization algorithm. Authors need to clarify this point clearly and discuss the potential real-time issue.

The reviewer is correct that the bi-level optimization can be a computational bottleneck, especially if the system involves a high number of degrees of freedom. As discussed at lines 422 to 457, computing the optimal control u involves integrating coupled ODEs due to the closed-loop LQ solutions, which is efficiently handled by standard solvers. For the high-level optimization, refining the initial guess may be sufficient in practice, as the optimization can be repeated at each control loop step (or even less frequently as we derive a feedback controller). Both levels can also be parallelized to speed up computations. We also provided some details in our previous response letter (see Comment 8). We have added the following statement in the same section to clarify this point (p.17, l.457): “[...] **Note that the computational time can be further reduced by leveraging parallel computing techniques.**”

Comment 7: The experimental validation lacks comparisons between the proposed approach and existing methods, such as [13]–[15].

As also mentioned in Comment 2 of our previous response letter, the existing algorithms in methods [13]-[15] could not be used due to the time-varying nature of our linearized dynamics. In fact, precisely this limitation of previous methods motivated us to develop the current method.

Comment 8: How about the real-time issue in the experiments? Although authors give an explanation to reduce the influence of the bi-level optimization during line 422 to 457, more detailed settings and results are lacked in experiments.

There was no real-time issue in our experiment. Using Matlab with Mex files and the BOBYQA solver from the NLOpt library, we were able to solve the problem within 10 ms. Details on BOBYQA tuning parameters in our experimental setup are provided at lines 439-457. Notably, the initial trust region radius was adapted based on the high-level cost magnitude to smooth variations in estimated cost weights. For example, when the high-level cost was small, BOBYQA did not need to initially explore a wide region of the parameters space. This heuristic rule is described in lines 449-453.

Answer to reviewers: Model predictive game control for personalized and targeted interactive assistance

We thank the reviewers and editors for their constructive comments, which we address point-to-point below. Our answers are provided in blue, and the corresponding modifications in the manuscript are also highlighted in blue. For convenience, we also provide a clean version of the manuscript without highlights.

Reviewer #2

Thank you for your reply. Although most of my concerns have been solved, there are still some major concerns remained

We thank the reviewer for the careful analysis of the submitted version and the detailed feedback.

Comment 1: The motivation for using finite horizon is still not strong enough by only introducing the results in [16]. Since this is the main strength of this paper compared with existing works. At least, comparisons between the approaches with finite and infinite horizons should be conducted. The reviewer has a great doubt about the disadvantages of infinite horizon.

Thank you for raising this important question. To clarify the differences between the finite- and infinite-horizon approaches and demonstrate the benefits of the proposed formulation, we provide an in-depth comparative analysis, presented after Comment 4. The main findings of this analysis show that the finite-horizon formulation: (i) avoids several ad hoc assumptions, (ii) achieves improved tracking performance, and (iii) maximizes the flexibility of the control strategy.

Comment 2: To make the work self-contained, authors are encouraged to conduct experiments for robustness analyses by using biased model parameters.

The robustness of our approach with respect to both dynamics and human modeling errors has been analyzed in our previous works [1]. These types of errors are inherently present in all our experiments, which further confirm the robustness of our approach. This information is now explicitly referenced in the Methods section (p.16, l.447):

“In simulation studies, we have shown that our method can robustly identify human cost parameters during 1-DoF movements with different initial guesses [2] and under noise/modeling errors in 2-DoF movements [1].”

Comment 3: The reviewer cares that whether the approach is just a simple extension of these previous works by allowing a tunable lambda or originally designed a novel framework for considering the modulatable human efforts.

The proposed finite-horizon formulation introduces more than just a tunable parameter—it enables a fundamentally more flexible interaction control strategy. Specifically, it not only provides superior control performance (as shown in Fig.1), but also supports the design of flexible and emerging effort sharing strategies (Fig.2), which are not feasible under infinite-horizon formulations. Equally important, our paper presents new experiments that i) validate the proposed framework, and ii) show that the robot can influence the role

adopted by the human partner (e.g. master-slave, slave-master, or intermediate cooperation modes). This provides tools for developing effective physical training strategies. Notably, such crucial investigations had not been reported in previous GT-based human-robot interaction studies [3–5].

Comment 4: Comparisons are needed for better explanation. Authors said that previous methods cannot be used for the time-variant linear system and thus cannot be compared. However, for previous approaches they just used different modeling approaches and still could be used for the same experimental tasks conducted in this paper. The reviewer expect to see that the proposed approach can achieve obviously better performances compared with existing approaches on the same human-robot collaboration tasks, especially on some points like using finite horizon and tunable human efforts.

We thank the reviewer for this valuable suggestion to provide a comparison with an infinite-horizon approach. As now discussed in the revised manuscript (at page 12, from line 300), we have clarified the theoretical and practical differences between the two formulations:

“[...] Existing DG algorithms for contact robots [3–5] assume an infinite planning horizon for the human agent, whereas humans naturally plan their movements over a finite horizon [6]. For its application, this setting requires a time-invariant system with null steady-state effort (as detailed at p.337 in [7]), conditions that are not necessarily satisfied in trajectory tracking or load-lifting assistance tasks. Therefore, implementing an infinite-horizon DG in our context necessitates several ad hoc assumptions, resulting in coarser approximations and reduced shared control abilities (see Supplementary Information). Moreover, previous infinite-horizon formulations can lead to the paradox that the human cost cannot be identified when the task is well performed and no error is observed [3, 4]. All these limitations are naturally resolved by the finite-horizon model predictive game (MPG) controller introduced in this paper, as demonstrated by our experiments and the simulations presented in the Supplementary Information.”

A detailed comparison between finite- and infinite-horizon approaches is provided below, which was included in the supplementary materials.

Finite vs infinite horizon

Summarizing the reviewer’s comments, three main points are raised concerning the novelty and distinctions of the finite-horizon DG formulation proposed in this paper, compared to the infinite-horizon formulations used in previous works e.g. [3, 4]:

1. *“The motivation for using a finite horizon is still not strong enough, [... and there is] great doubt about the disadvantages of infinite horizon.”*
2. *[It is unclear] “whether the approach is just a simple extension of previous infinite horizon works, allowing a tunable effort sharing through λ , or an originally designed framework for personalized and flexible human-robot effort sharing.”*
3. *“Authors said that previous methods cannot be used for the time-variant linear system and thus cannot be compared. However, for previous approaches they just used different modeling approaches and still could be used for the same experimental tasks.”*

To address these concerns, we analyzed through simulations the differences between finite- and infinite-horizon formulations of DG-based interaction. This analysis, described below and reported as Supplementary Information, reveals that:

- In our context of tracking with nonlinear dynamics, implementing the infinite-horizon game requires an *ad hoc* hypothesis to remove the affine term from the dynamics, making the two approaches fundamentally different (addressing points 1 and 3).

- The finite-horizon approach results in better tracking performance than the infinite-horizon formulation, as shown in Fig.1 (addressing points 1 & 3).
- The finite-horizon approach enables flexible tuning of the optimal effort sharing between agents while (i) ensuring that the robot adapts to each individual in real time, and (ii) supporting human-robot co-adaptation. Both features were demonstrated in our experiments, and in the simulations, see Fig.2 (addressing points 2 & 3).
- In contrast, the infinite-horizon approach constrains the effort distribution between agents to a fixed ratio at the dynamics level (Fig.3), preventing real-time individualization and spontaneously emerging co-adaptation (addressing points 2 & 3).
- In conclusion, our finite-horizon methodology for interaction-modulation and effort-sharing strategies in human-robot interaction outperforms previous infinite-horizon approaches in both performance (points 1 & 3) and flexibility (points 2 & 3).

Below, we analyze the main differences between a finite- and infinite-horizon DG in human-robot interaction to understand how the choice of horizon affects the game formulation, the resulting Nash equilibrium and the control performance. This analysis is conducted in a simple trajectory tracking simulation with wrist flexion/extension movements up and down, consistent with our experiments.

Preliminary considerations

In the standard infinite-horizon formulation, the state and control matrices must be time-invariant and the motor controls of the human and robot agents (u_h and u_r) are defined through deviations, This enables the application of the standard DG solution (see pp. 336–7 in [7]). This formulation implicitly assumes that the equilibrium torques are zero, i.e. there is no steady force to counteract. These assumptions hold in setups such as that of [3], where interaction was constrained to a horizontal plane and the equilibrium was unaffected by gravity. In that context, the control represented pure feedback terms around a zero-torque equilibrium, and both the shared state and agents cost functions could be expressed as deviations from this equilibrium.

In contrast, our setup involves varying gravitational terms, so that even at equilibrium the system requires a nonzero steady torque. Modeling our scenario with an infinite-horizon approach therefore requires (i) linearizing the dynamics around a fixed point to obtain time-invariant matrices, and (ii) an *a priori* assumption about how the agents share this steady torque. This assumption is implemented through the parameter γ (introduced below) that operates similarly to λ used in the cost functions, but at the dynamics level, as described in the Interaction dynamics section. Both the linearization point and the steady torque sharing are *ad hoc* choices to make the infinite horizon DG settings applicable.

Moreover, in such purely feedback formulations, when the task is well performed —i.e. when the tracking error $\xi = 0$ and $\mu_h = 0$ — the human cost function becomes unobservable. This is explicitly acknowledged in [3] (at p. 38, by Eq. (10)): “[...] *the partners’ controllers and cost functions converge to the correct values if ξ is persistently exciting*”. Hence, **this previous formulation required task deviations to observe the human cost function, which is feasible in reaching tasks (as the error needs to vanish only at the end of the movement) but not in continuous tracking.**

Our finite-horizon approach overcomes these limitations. It (i) can handle time-varying state and control matrices, naturally through the backward integration of the ODEs implementing the DG, and (ii) directly optimizes u_h and u_r considering both feedback and the feedforward components that emerge from the task dynamics. As a first consequence, **the agents can autonomously determine how to share the efforts required to accomplish the task in real time, without requiring an ad hoc parameter γ .** Second, **the agents’ control is not null even when $\xi = 0$, enabling the human strategy to remain observable when the task is perfectly executed.**

Below, we describe how these considerations impact the formulation of the problem, the tracking performance, and the contributions of the agents for different human behaviors.

Interaction dynamics

We consider the coupled human-robot-load system:

$$\mathbf{f}(\mathbf{x}, u, t) = \begin{bmatrix} \dot{q} \\ [u_h + u_r - \mathcal{G}(q) - \mathcal{D}\dot{q}] / \mathcal{I} \end{bmatrix}. \quad (1)$$

where q is the position, u_r and u_h are the robot and human applied torques respectively, \mathcal{I} , \mathcal{D} and \mathcal{G} are the inertia, damping and the (nonlinear) gravitational term, respectively. Linearisation around some fixed desired trajectory-control pair (\mathbf{x}_d, u_d) for infinite-horizon setup yields:

$$\begin{aligned} \dot{\boldsymbol{\xi}}(t) &= \mathbf{A}\boldsymbol{\xi}(t) + \mathbf{B}[\mu_r(t) + \mu_h(t)], \quad \boldsymbol{\xi} = \mathbf{x} - \mathbf{x}_d, \quad \mathbf{x} = \begin{bmatrix} q \\ \dot{q} \end{bmatrix}, \quad \mathbf{x}_d = \begin{bmatrix} q_d \\ \dot{q}_d \end{bmatrix}, \\ \mathbf{A} &\triangleq \frac{\partial \mathbf{f}}{\partial \mathbf{x}}|_{[\mathbf{x}_d(\tau+\Delta_p), u_d(\tau+\Delta_p)]}, \quad \mathbf{B} \triangleq \frac{\partial \mathbf{f}}{\partial u}|_{[\mathbf{x}_d(\tau+\Delta_p), u_d(\tau+\Delta_p)]}. \end{aligned} \quad (2)$$

where τ is the actual time, Δ_p is the planing horizon fixed in the finite-horizon approach, and the control inputs are defined as $\mu_r = u_r - \gamma u_d$ and $\mu_h = u_h - (1 - \gamma)u_d$. As previously mentioned, the infinite-horizon imposes to linearize around a fixed point, which we define at $t = \tau + \Delta_p$, where $\Delta_p = 0.5$ s is the planning horizon of the finite-horizon formulation. Hence, although we will apply DG on an infinite horizon, the target still corresponds to some upcoming state that the controls need to fulfill.

The formulation of the finite-horizon approach remains unchanged as presented in our method.

MPG infinite horizon controller implementation

The human and robot optimize their respective cost functions in the infinite horizon setup:

$$J_h = \frac{1}{2} \int_{\tau}^{\infty} \boldsymbol{\xi}^{\top} \mathbf{Q}_h \boldsymbol{\xi} + R_h \mu_h^2 dt, \quad (3)$$

$$J_r = \frac{1}{2} \int_{\tau}^{\infty} \boldsymbol{\xi}^{\top} \mathbf{Q}_r \boldsymbol{\xi} + R_r \mu_r^2 + R_{rh} \mu_h^2 dt. \quad (4)$$

For this comparison, we set $R_h = 1$, $R_r = 0.5 + (1 - \lambda)$, and $R_{rh} = 0.5 + \lambda$ similarly to our finite-horizon setup. Note that the cross term R_{rh} lacks physical meaning here, as minimizing μ_h does not correspond to a physically interpretable interaction variable. This highlights another limitation of the infinite-horizon framework: it cannot represent cooperative terms that correspond to physically meaningful shared effort.

The optimal control is obtained by solving the coupled Riccati-like equations (see p. 337, Eq. (6.87) in [7]), using Newton-Kleinman algorithm at each timestep with the same MPG approach as for the finite horizon:

$$\begin{cases} \mathbf{F}^{\top} \mathbf{P}_h + \mathbf{P}_h \mathbf{F} + \mathbf{P}_h \mathbf{B} R_h^{-1} \mathbf{B}^{\top} \mathbf{P}_h + \mathbf{Q}_h = 0, \\ \mathbf{F}^{\top} \mathbf{P}_r + \mathbf{P}_r \mathbf{F} + \mathbf{P}_r \mathbf{B} R_r^{-1} \mathbf{B}^{\top} \mathbf{P}_r + \mathbf{P}_h \mathbf{B} R_h^{-1} R_{rh} R_h^{-1} \mathbf{B}^{\top} \mathbf{P}_h + \mathbf{Q}_r = 0, \\ \mathbf{F} = \mathbf{A} - \mathbf{B} R_r^{-1} \mathbf{B}^{\top} \mathbf{P}_r - \mathbf{B} R_h^{-1} \mathbf{B}^{\top} \mathbf{P}_h, \end{cases} \quad (5)$$

and the optimal feedback commands are:

$$\mu_i(\tau) = -R_i^{-1} \mathbf{B}^{\top} \mathbf{P}_i \boldsymbol{\xi}, \quad \mu_h(\tau) = -R_h^{-1} \mathbf{B}^{\top} \mathbf{P}_h \boldsymbol{\xi}. \quad (6)$$

Simulation settings

We simulated the controllers with the following parameters: $\mathcal{I} = 0.1 \text{ kg m}^2$, $\mathcal{D} = 0.1 \text{ kg m}^2/\text{s}$, $m = 1 \text{ kg}$, $l = 0.1 \text{ m}$, and $g = 9.81 \text{ m/s}^2$. The nominal control input was:

$$u_d(t) = \mathcal{I} \ddot{q}_d(t) + \mathcal{D} \dot{q}_d(t) + mgl \sin[q_d(t)], \quad (7)$$

with the target trajectory defined as:

$$q_d(t) = \frac{\pi}{4} [1 + \cos(\pi t)], \quad (8)$$

the initial position and velocity were set at $q(0) = \pi/4$, $\dot{q}(0) = 0$, and the cost matrices were set as $\mathbf{Q}_r = \mathbf{Q}_{max}/2$ and $\mathbf{Q}_h = \zeta \mathbf{Q}_{max}$, $0 \leq \zeta \leq 1$, with $\mathbf{Q}_{max} = \text{diag}(200, 1)$.

Results

Tracking performance

Fig. 1 A,B illustrate how the finite- and infinite-horizon DG controllers track the reference trajectory (for the direct DG problem with known costs). The mean position and velocity errors shown in Panels C and D indicate that the infinite-horizon controller exhibits substantially higher errors than the finite-horizon controller (42.5% in position and 33.3% in velocity). This degradation in performance is mainly due to the linearization approximation, where at each time step the infinite-horizon approach assumes a fixed target at $t = \tau + \Delta_p$ to yield time-invariant state matrices, whereas in the finite horizon approach the linearization is performed along the desired trajectory over the future finite horizon $t \in [\tau, \tau + \Delta_p]$. These results highlight the importance of the finite-horizon formulation for accurately tracking both the desired trajectory and the dynamics.

Figure 1: Tracking performance. **A.** Position evolution over time. **B.** Velocity evolution over time. **C.** Mean position error with standard error over 5 seconds of tracking for finite- and infinite-horizon DG controllers. **D.** Mean velocity error with standard error over 5 seconds of tracking for finite- and infinite-horizon DG controllers.

Effort sharing comparison

We further compared the effort sharing between the finite- and infinite-horizon formulations. Fig 2 shows the mean applied torque of the robot and human, during 5 seconds of the tracking task, for different values

of the human cost matrix \mathbf{Q}_h and the parameter λ . We observe that varying either \mathbf{Q}_h or λ has a negligible effect in the effort sharing with the infinite-horizon approach, unlike in the finite-horizon approach. This difference arises because, in the infinite horizon case, the total applied effort consists of a shared feedback correction term and a fixed feedforward component. The latter compensates for the steady torque, which are predominant, resulting in a fixed effort distribution between agents when the task is well performed.

Figure 2: Mean human and robot effort over 5 s of tracking as a function of $\mathbf{Q}_h/\mathbf{Q}_{max}$ for different λ values and $\gamma = 1$.

Although varying γ can redistribute the effort between agents (Fig. 3), it rigidly dictates the effort-sharing pattern. This means that the infinite-horizon formulation prevents any individualization or significant co-adaptation, since variations in \mathbf{Q}_h have a negligible impact when the task is well executed with $\xi \simeq 0$. Thus, the infinite-horizon approach loses a principal advantage of the DG control formulation: its dynamic adaptability.

Figure 3: Mean human and robot effort over 5 seconds of tracking as a function of $\mathbf{Q}_h/\mathbf{Q}_{max}$ for different γ values, for $\lambda = 0.5$.

In contrast, the finite-horizon formulation incorporates the evolving task dynamics, allowing both \mathbf{Q}_h and λ to influence the resulting control effort. As shown in Fig. 2, this results in distinct variations in human and robot applied torques as these parameters change, enabling flexible and personalized effort sharing. Moreover, our experiments demonstrate that this mechanism allows the robot to shape the interaction relationship with the human user, an ability that could, for example, be used to optimize physical training protocols.

In conclusion, the finite-horizon is more flexible and exhibits unique properties to model the co-adaptation of two intelligent agents sharing effort in a load-carrying tracking task.

References

- [1] A. Hafs, D. Verdel, O. Bruneau, and B. Berret, “Game-Theoretic Interaction Control for Assistive Exoskeletons: a 2-DOF Simulation Study,” in IFAC 2025 - Joint 10th IFAC Symposium on Mechatronic Systems and 14th Symposium on Robotics, Paris, France, Jul. 2025. [Online]. Available: <https://hal.science/hal-05063077>
- [2] A. Hafs, D. Verdel, E. Burdet, O. Bruneau, and B. Berret, “A finite-horizon inverse differential game approach for optimal trajectory-tracking assistance with a wrist exoskeleton,” in 2024 10th IEEE RAS/EMBS International Conference for Biomedical Robotics and Biomechanics (BioRob), 2024, pp. 450–456.
- [3] Y. Li, G. Carboni, F. Gonzalez, D. Campolo, and E. Burdet, “Differential game theory for versatile physical human–robot interaction,” Nature Machine Intelligence, vol. 1, no. 11, p. 36–43, Jan. 2019.
- [4] S. Musić and S. Hirche, “Haptic shared control for human-robot collaboration: A game-theoretical approach,” IFAC-PapersOnLine, vol. 53, no. 2, p. 10216–10222, Jan. 2020.
- [5] L. Pezeshki, H. Sadeghian, M. Keshmiri, X. Chen, and S. Haddadin, “Cooperative assist-as-needed control for robotic rehabilitation: A two-player game approach,” IEEE Robotics and Automation Letters, vol. 8, no. 5, p. 2852–2859, May 2023.
- [6] L. Bashford, D. Kobak, J. Diedrichsen, and C. Mehring, “Motor skill learning decreases movement variability and increases planning horizon,” Journal of Neurophysiology, vol. 127, no. 4, p. 995–1006, Apr. 2022. [Online]. Available: <http://dx.doi.org/10.1152/jn.00631.2020>
- [7] T. Başar and G. J. Olsder, Dynamic Noncooperative Game Theory, 2nd Edition, ser. Classics in Applied Mathematics. Society for Industrial and Applied Mathematics, Jan. 1998. [Online]. Available: <https://epubs.siam.org/doi/book/10.1137/1.9781611971132>